# Photosynthetic sea slugs induce protective changes to the light reactions of the chloroplasts they steal from algae

**Vesa Havurinne, Esa Tyystjärvi\***

University of Turku, Department of Biochemistry / Molecular Plant Biology, Turku, Finland

**Abstract** Sacoglossan sea slugs are able to maintain functional chloroplasts inside their own cells, and mechanisms that allow preservation of the chloroplasts are unknown. We found that the slug *Elysia timida* induces changes to the photosynthetic light reactions of the chloroplasts it steals from the alga *Acetabularia acetabulum*. Working with a large continuous laboratory culture of both the slugs (>500 individuals) and their prey algae, we show that the plastoquinone pool of slug chloroplasts remains oxidized, which can suppress reactive oxygen species formation. Slug chloroplasts also rapidly build up a strong proton-motive force upon a dark-to-light transition, which helps them to rapidly switch on photoprotective non-photochemical quenching of excitation energy. Finally, our results suggest that chloroplasts inside *E. timida* rely on oxygen-dependent electron sinks during rapid changes in light intensity. These photoprotective mechanisms are expected to contribute to the long-term functionality of the chloroplasts inside the slugs.

**\*For correspondence:**
esatyy@utu.fi

**Competing interests:** The authors declare that no competing interests exist.

## Introduction

The sea slug *Elysia timida* is capable of stealing chloroplasts from its algal prey (*Figure 1*). Once stolen, the chloroplasts, now termed kleptoplasts, remain functional inside the slug's cells for several weeks, essentially creating a photosynthetic slug. The only animals capable of this phenomenon are some marine flatworms (*Van Steenkiste et al., 2019*) and sea slugs belonging to the Sacoglossan clade (*Rumpho et al., 2011*; *de Vries et al., 2014*). Despite decades of research, there is still no consensus about the molecular mechanisms that allow the slugs to discriminate other cellular components of the algae and only incorporate the chloroplasts inside their own cells, or how the slugs maintain the chloroplasts functional for times that defy current paradigms of photosynthesis. Also the question whether the slugs in fact get a real nutritional benefit from the photosynthates produced by the stolen chloroplasts, is still being debated (*Cartaxana et al., 2017*; *Rauch et al., 2017*).

One of the main problems that kleptoplasts face is light-induced damage to both photosystems. Photoinhibition of Photosystem II (PSII) takes place at all light intensities and photosynthetic organisms have developed an efficient PSII repair cycle to counteract it (*Tyystjärvi, 2013*). Unlike higher plants (*Järvi et al., 2015*), the chloroplast genomes of all algal species involved in long-term kleptoplasty encode FtsH, a protease involved in PSII repair cycle (*de Vries et al., 2013*). However, out of all prey algae species of photosynthetic sea slugs, only in *Vaucheria litorea*, the prey alga of *Elysia chlorotica*, the chloroplast-encoded FtsH contains the critical M41 metalloprotease domain required for degradation of the D1 protein during PSII repair (*Christa et al., 2018*). Photoinhibition of Photosystem I (PSI) occurs especially during rapid changes in light intensity (*Tikkanen and Grebe, 2018*) and should cause problems in isolated chloroplasts in the long run. In addition to the specific inhibition mechanisms of the photosystems, unspecific damage caused by reactive oxygen species (ROS) (*Khorobrykh et al., 2020*) is expected to deteriorate an isolated chloroplast.

**eLife digest** Plants, algae and a few other organisms rely on a process known as photosynthesis to fuel themselves, as they can harness cellular structures called chloroplasts to convert light into usable energy. Animals typically lack chloroplasts, making them unable to use photosynthesis to power themselves. The sea slug *Elysia timida*, however, can steal whole chloroplasts from the cells of the algae it consumes: the stolen structures then become part of the cells in the gut of the slug, allowing the animal to gain energy from sunlight.

Once they are in the digestive system of the slug, the chloroplasts survive and keep working for longer than expected. Indeed, these structures are often harmed as a side effect of photosynthesis, but the sea slug does not have the right genes to help repair this damage. In addition, conditions inside animal cells are widely different to the ones found inside algae and plants. It is not clear then how the sea slug extends the lifespan of its chloroplasts by preventing damage caused by sunlight.

To investigate this question, Havurinne and Tyystjärvi compared photosynthesis in sea slugs and the algae they eat. A range of methods, including measuring fluorescence from the chloroplasts, was used: this revealed that the slug changes the inside of the stolen chloroplasts, making them more resistant to damage.

First, when exposed to light the stolen chloroplasts can quickly switch on a mechanism that dissipates light energy to heat, which is less damaging. Second, a molecule that serves as an intermediate during photosynthesis is kept in a 'safe' state which prevents it from creating harmful compounds. And finally, additional safeguard molecules 'deactivate' compounds that could otherwise mediate damaging reactions. Overall, these measures may reduce the efficiency of the chloroplasts but allow them to keep working for much longer.

Early chloroplasts were probably independent bacteria that were captured and 'domesticated' by other cells for their ability to extract energy from the sun. Photosynthesizing sea slugs therefore provide an interesting way to understand some of the challenges of early life. The work by Havurinne and Tyystjärvi may also reveal new ways to harness biological processes such as photosynthesis for energy production in other contexts.

Photoprotective mechanisms counteract photodamage. Recent efforts have advanced our understanding of photoprotection in kleptoplasts (*Christa et al., 2018*; *Cartaxana et al., 2019*). It has been shown that kleptoplasts of *E. timida* do retain the capacity to induce physiological photoprotection mechanisms similar to the ones in the prey green alga *Acetabularia acetabulum* (hereafter *Acetabularia*) (*Christa et al., 2018*). The most studied mechanism is the 'energy-dependent' qE component of non-photochemical quenching of excitation energy (NPQ). qE is triggered through acidification of the thylakoid lumen by protons pumped by the photosynthetic electron transfer chain. The xanthophyll cycle enhances qE (*Ruban and Horton, 1999*; *Papageorgiou GC, 2014*) but there have been contrasting reports on the capability of *E. timida* to maintain a highly functional xanthophyll cycle if the slugs are not fed with fresh algae (i.e. starved) and about the effect of NPQ on kleptoplast longevity (*Christa et al., 2018*; *Cartaxana et al., 2019*). Although advancing our understanding of the mechanisms of kleptoplast longevity, these recent publications underline the trend of contradictory results that has been going on for a long time.

There are reports of continuous husbandry of photosynthetic sea slugs, mainly *E. timida* (*Schmitt et al., 2014*) and *E. chlorotica* (*Rumpho et al., 2011*), but still today most research is conducted on animals caught from the wild. We have grown the sea slug *E. timida* and its prey *Acetabularia* in our lab for several years (*Figure 1*). As suggested by *Schmitt et al., 2014*, *E. timida* is an attractive model organism for photosynthetic sea slugs because it is easy to culture with relatively low costs (*Figure 1E*). A constant supply of slugs has opened a plethora of experimental setups yet to be tested, one of the more exciting ones being the case of red morphotypes of both *E. timida* and *Acetabularia* (*Figure 1C,D*). Red morphotypes of *E. timida* and *Acetabularia* were first described by *González-Wangüemert et al., 2006* and later shown to be due to accumulation of an unidentified carotenoid during cold/high-light acclimation of the algae that were then eaten by *E. timida* (*Costa et al., 2012*). The red morphotypes provide a visual proof that the characteristics of the

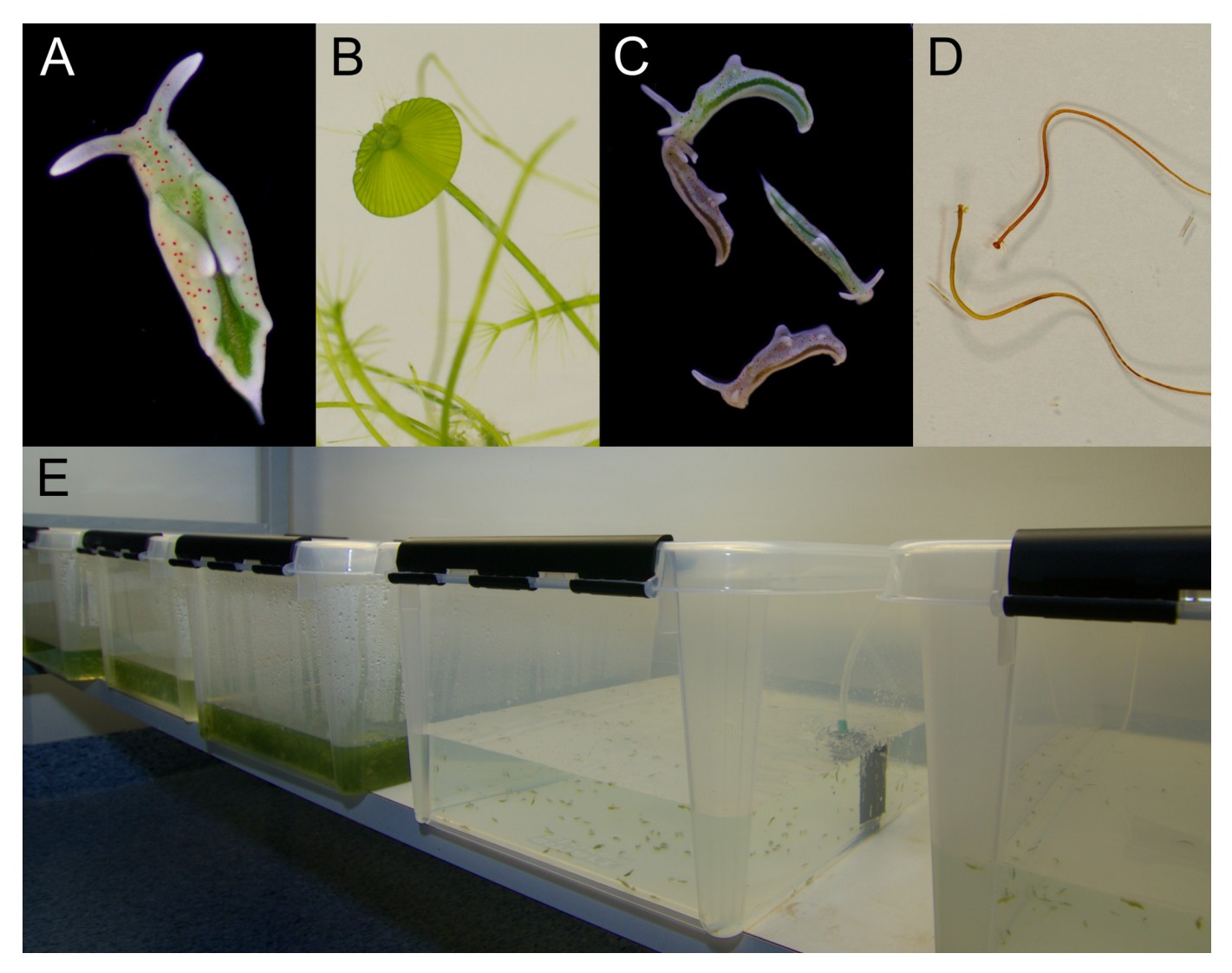

**Figure 1.** Laboratory cultures of the photosynthetic sea slug *E. timida* and its prey alga *Acetabularia*. (**A**) A freshly fed adult *E. timida* individual. The length of an adult slug in our culture conditions is approximately 6 mm. (**B**) The giant-celled green alga *Acetabularia*. The cap-like structures are the site of gamete maturation and serve as indicators of the end of the vegetative growth phase of individual *Acetabularia* cells. The length of an individual *Acetabularia* cell at the end of the vegetative phase can reach 50 mm. (**C**) The red morphotype *E. timida* can be induced by feeding it with red morphotype *Acetabularia*. Green morphotype *E. timida* individuals are also shown for reference. (**D**) Red morphotype *Acetabularia* can be induced by subjecting the cells to cold temperature and high light (see 'Materials and methods' for details). (**E**) *E. timida* and *Acetabularia* can be cultured in transparent plastic tanks. The two tanks in the foreground are *E. timida* tanks and the three other tanks contain *Acetabularia* cultures.

kleptoplasts inside *E. timida* can be modified by acclimating their feedstock to different environmental conditions.

We optimized a new set of biophysical methods to study photosynthesis in the sea slugs and found differences in photosynthetic electron transfer reactions between *E. timida* and *Acetabularia* grown in varying culture conditions. The most dramatic differences between the slugs and their prey were noticed in PSII electron transfer of the red morphotype *E. timida* (*Figure 1C*) and *Acetabularia* (*Figure 1D*). In addition to measuring chlorophyll *a* fluorescence decay kinetics, we also measured fluorescence induction kinetics, PSI electron transfer and formation of proton-motive force during dark to light transition. Our results suggest that dark reduction of the plastoquinone (PQ) pool, a reserve of central electron carriers of the photosynthetic electron transfer chain, is weak in the slugs compared to the algae, and that a strong build-up of proton-motive force is likely linked to higher

levels of NPQ in kleptoplasts. It is also clear that PSI utilizes oxygen-sensitive electron sinks in both the slugs and the algae, and this sink protects the photosynthetic apparatus from light-induced damage in *E. timida*.

## Results

### Non-photochemical reduction of the chloroplast electron transfer chain is inefficient in *E. timida*

We estimated reoxidation kinetics of the first stable electron acceptor of PSII, $Q_A^-$, from dark acclimated *E. timida* and *Acetabularia* by measuring the decay of chlorophyll *a* fluorescence yield after a single turnover flash (*Figure 2*). In the green morphotypes of the slugs and the algae, $Q_A^-$ reoxidation kinetics were similar between the two species in aerobic conditions both in the absence and presence of 3-(3, 4-dichlorophenyl)−1, 1-dimethylurea (DCMU), an inhibitor of PSII electron transfer (*Figure 2A,B*). This indicates that electron transfer within PSII functions in the same way in both species. In anaerobic conditions, fluorescence decay was slower than in aerobic conditions in both species (*Figure 2C*), showing that the environment of the slug kleptoplasts normally remains aerobic in the dark even in the presence of slug respiration. Anaerobicity slowed the fluorescence decay less in *E. timida* than in *Acetabularia*, especially during the fast (~300–500 µs) and middle (~5–15 ms) phases of fluorescence decay in anaerobic conditions (*Figure 2C*). Decay of fluorescence is slow in anaerobic conditions likely because dark reduction of the electron transfer chain, specifically the PQ pool, hinders $Q_A^-$ reoxidation (*de Wijn and van Gorkom, 2001*; *Oja et al., 2011*; *Deák et al., 2014*; *Krishna et al., 2019*). This suggests that non-photochemical reduction of the electron transfer chain during the dark acclimation period is less efficient in the slug than in the alga.

The decay of fluorescence after a single turnover flash followed strong wave like kinetics in the red morphotype *Acetabularia*, with a large undershoot below the dark-acclimated minimum fluorescence level, while there was no sign of such kinetics in the red morphotype *E. timida* (*Figure 2D*, see also *Figure 1C,D* for images of the red morphotypes and 'Materials and methods' for their preparation). The wave phenomenon has been characterized in detail in several species of cyanobacteria, where anaerobic conditions in the dark are enough for its induction (*Wang et al., 2012*; *Deák et al., 2014*; *Ermakova et al., 2016*). According to *Deák et al., 2014*, anaerobic conditions cause a highly reduced PQ pool through respiratory electron donation, mainly from NAD(P)H. This reduction is mediated by the NAD(P)H-dehydrogenase (NDH). Taken together, the results shown in *Figure 2C,D* indicate that the dark reduction of the chloroplast's electron transfer chain is not as strong in *E. timida* as it is in *Acetabularia*.

### Full photochemical reduction of the electron transfer chain during a dark-to-light transition is delayed in *E. timida*

In order to investigate whether the alterations in the dark reduction of the electron transfer chain in kleptoplasts lead to differences in electron transfer reactions during continuous illumination, we measured chlorophyll *a* fluorescence induction kinetics from both *E. timida* and *Acetabularia* (*Figure 3*). Briefly, fluorescence rise during the first ~ 1 s of continuous illumination of a photosynthetic sample can be divided into distinct phases, denoted as O-J-I-P, when plotted on a logarithmic time scale (see *Figure 3A*). Alterations in the magnitude and time requirements of these phases are indicative of changes in different parts of the photosynthetic electron transfer chain (*Strasserf and Srivastava, 1995*; *Strasser et al., 2004*; *Kalaji et al., 2014*).

Fluorescence induction measurements in aerobic conditions revealed that in green *E. timida* individuals maximum fluorescence (P phase of OJIP fluorescence rise kinetics) was reached ~ 300 ms later than in green *Acetabularia* (*Figure 3A*). To investigate whether the elongated time requirement to reach maximum fluorescence is caused by light attenuation in the slug tissue, we tested the effect of different intensities of the light pulse to the fluorescence transient in *E. timida*. In the tested range, the intensity of the pulse did affect the O-J-I phases but not the time of the P phase in aerobic conditions (*Figure 3—figure supplement 1*). This suggests that the ~ 300 ms delay of the P phase in *E. timida* is of physiological origin, and once again an indicator of differing redox poises of the chloroplasts between *E. timida* and *Acetabularia*.

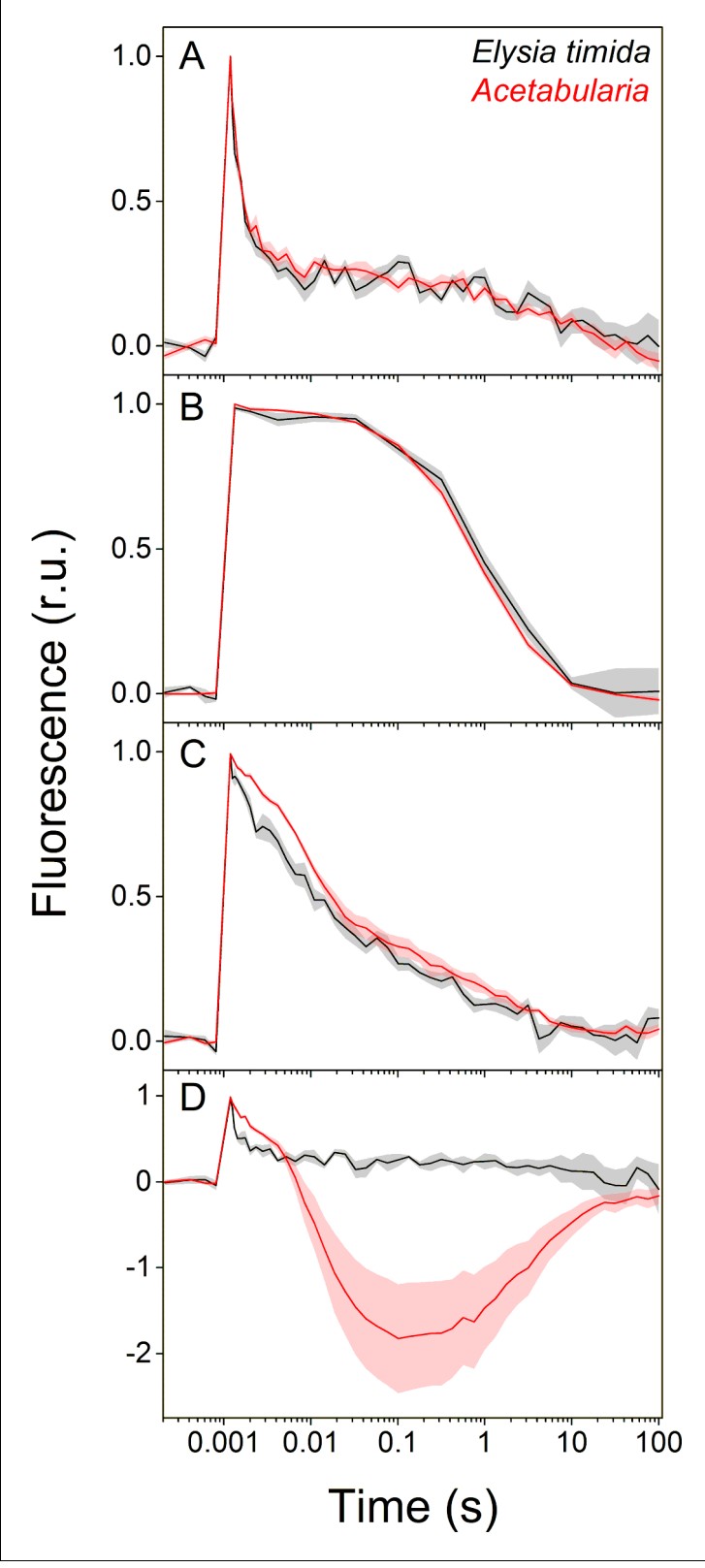

**Figure 2.** Differences in $Q_A^-$ reoxidation between *E. timida* (black) and *Acetabularia* (red) suggest differences in the redox state of the PQ pool after dark acclimation. (**A–C**) Chlorophyll *a* fluorescence yield decay after a single turnover flash in aerobic conditions without any inhibitors in regular, green morphotypes of *E. timida* and *Acetabularia* (**A**), in aerobic conditions in the presence of 10 µM DCMU (**B**), and in anaerobic conditions, achieved

*Figure 2 continued on next page*

*Figure 2 continued*

by a combination of glucose oxidase (eight units/ml), glucose (6 mM) and catalase (800 units/ml), in the absence of inhibitors (C). See *Figure 2—figure supplement 1A* for details on the anaerobic conditions. (D) Chlorophyll fluorescence decay measured from the red morphotypes of *E. timida* and *Acetabularia* in aerobic conditions without any inhibitors. Fluorescence traces were double normalized to their respective minimum (measured prior to the single turnover flash), and maximum fluorescence levels. Curves in (A) are averages from 7 (*E. timida*) and 5 (*Acetabularia*) biological replicates, 4 and 5 in (B), and 5 and 5 in (C–D), respectively. The shaded areas around the curves represent SE. All *E. timida* data are from individuals taken straight from the feeding tanks without an overnight starvation period. See *Figure 2—source data 1* for original data.

The online version of this article includes the following source data and figure supplement(s) for figure 2:

**Source data 1.** Source data of chlorophyll fluorescence measurements shown in *Figure 2*.

**Figure supplement 1.** Oxygen consumption by the glucose oxidase system (eight units/ml glucose oxidase, 6 mM glucose and 800 units/ml catalase) in room temperature.

---

We witnessed a slightly slower O-J phase in DCMU-treated *E. timida* individuals (~10 ms to reach J) than in DCMU treated *Acetabularia* (~6 ms) (*Figure 3B*). The O-J phase is considered to represent the reduction of $Q_A$ to $Q_A^-$, and it is indicative of the amount of excitation energy reaching PSII, that is functional absorption cross-section of the PSII light harvesting antennae (*Kalaji et al., 2014*). The J-I-P phases are nullified when DCMU is introduced into the sample, as forward electron transfer from $Q_A^-$ is blocked (*Kodru et al., 2015*), which makes the O-J phase highly distinguishable. While it is conceivable that the slower O-J fluorescence rise (*Figure 3B*) indicates a decrease in the functional absorption cross-section of PSII in the slug cells, artefactual changes to the fluorescence signal by the different optical properties of the slugs and the algae cannot be ruled out as an explanation for the minute difference (~4 ms) in the O-J phase based on the current data. Comparison of the differing photosynthetic matrices between the slugs and the algae, and their effects on the photosynthetic parameters derived from fluorescence measurements have been thoroughly discussed in *Cruz et al., 2013* and *Serôdio et al., 2014*. We cannot completely rule out the possibility that DCMU does not block all PSII units of the kleptoplasts but find this unlikely, as addition of DCMU caused a similar change in the form of the OJIP curve in both slugs and algae (*Figure 3A and B*).

In anaerobic conditions the OJIP transient behaved in a manner that can be explained by a highly reduced electron transfer chain (*Figure 3C*). This blockage seems to affect the electron transfer more in *Acetabularia* than in *E. timida*, that is the J-I-P phases are more pronounced in *E. timida*, supporting the earlier suggestion derived from *Figure 2C*, that in anaerobic conditions electron transfer from $Q_A^-$ to $Q_B$ and PQ pool is faster in *E. timida* than in *Acetabularia*.

## Chloroplasts in *E. timida* exhibit strong build-up of proton-motive force in the light

To inspect intricate differences in proton-motive force formation between *E. timida* and *Acetabularia*, we measured electrochromic shift (ECS) from dark acclimated individuals of both species during a strong, continuous light pulse (*Figure 4A*). According to the ECS data, proton-motive force of *Acetabularia* dissipates to a steady level after the initial spike in thylakoid membrane energization, whereas in *E. timida* there is a clear build-up of proton-motive force after a slight relaxation following the initial spike. ECS was also measured from *E. timida* individuals that were devoid of kleptoplasts, and the slug tissue itself was found not to cause any inherent distortions to the ECS signal (*Figure 4—figure supplement 1*).

## *E. timida* and *Acetabularia* utilize oxygen-dependent electron sinks from PSI

We utilized a nearly identical protocol as *Shimakawa et al., 2019* to measure redox kinetics of P700, the reaction center chlorophyll of PSI, in dark acclimated *E. timida* and *Acetabularia* during dark-to-light transition (*Figure 4B–D*, see also *Figure 4—figure supplement 1C* for P700 redox kinetics measurements from *E. timida* individuals without any kleptoplasts). In aerobic conditions, *Acetabularia* P700 redox kinetics during a high-light pulse followed the scheme where P700 is first strongly oxidized due to PSI electron donation to downstream electron acceptors such as ferredoxin, and re-reduced by electrons from the upstream electron transfer chain (*Figure 4B*). Finally, oxidation

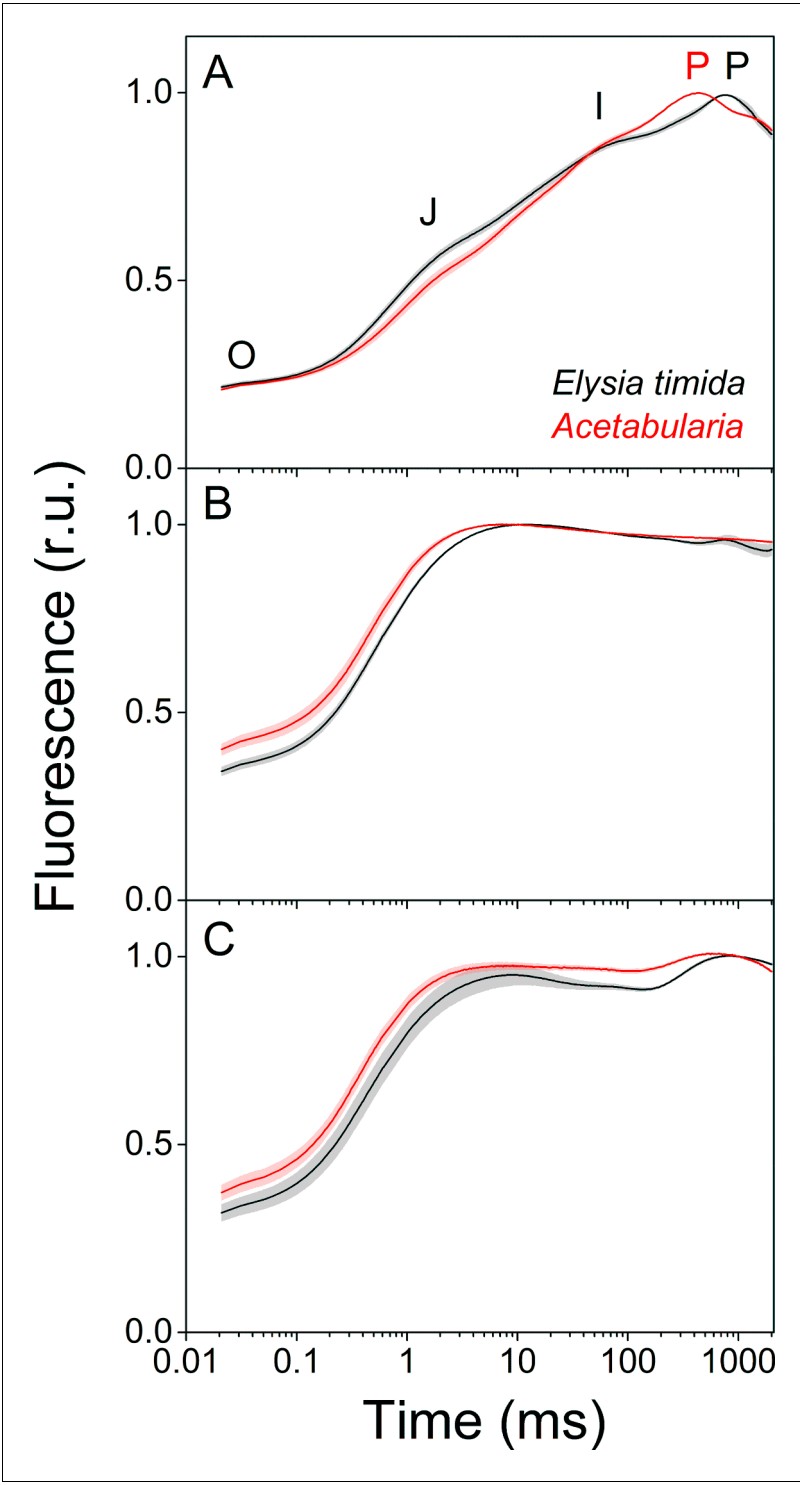

**Figure 3.** Fluorescence induction kinetics during dark-to-light transition suggest differences in full photochemical reduction of the PQ pool between *E. timida* (black) and *Acetabularia* (red). (A–C) Multiphase chlorophyll *a* fluorescence induction transient (OJIP) measured from dark acclimated *E. timida* and *Acetabularia* in aerobic conditions without any inhibitors (A), in the presence of 10 µM DCMU (B), and in anaerobic conditions, achieved by a combination of glucose oxidase (eight units/ml), glucose (6 mM) and catalase (800 units/ml), without any inhibitors (C). Fluorescence traces were normalized to their respective maximum fluorescence levels. For unnormalized data, see *Figure 3—figure supplement 1B*. Curves in (A) are averages from 10 (*E. timida*) and 12 (*Acetabularia*) biological replicates, 10 and 9 in (B), and 13 and 11 in (C), respectively. The shaded areas around
*Figure 3 continued on next page*

*Figure 3 continued*

the curves represent SE. All *E. timida* data are from individuals taken straight from the feeding tanks without an overnight starvation period. See *Figure 3—source data 1* for original data.

The online version of this article includes the following source data and figure supplement(s) for figure 3:

**Source data 1.** Source data of OJIP measurements shown in *Figure 3*.

**Figure supplement 1.** Technical considerations of the OJIP fluorescence induction measurements.

---

is resumed by alternative electron acceptors of PSI, most likely FLVs, as they have been shown to exist in all groups of photosynthetic organisms except angiosperms and certain species belonging to the red-algal lineage (*Allahverdiyeva et al., 2015*; *Ilík et al., 2017*; *Shimakawa et al., 2019*). P700 redox kinetics in *E. timida* and *Acetabularia* in aerobic conditions were similar in terms of the overall shape of the curve, but the re-oxidation phase after ~ 600 ms was dampened in *E. timida* (*Figure 4B*). In the presence of DCMU, P700 remained oxidized throughout the pulse in both species (*Figure 4C*). Data concerning non-photochemical reduction of P700$^+$ after the light pulse in the presence of DCMU are heavily affected by the normalization process of the signal in *E. timida*, and therefore minor fluctuations of the signal cannot be taken as evidence for physiological phehomena. The DCMU data does, however, show that the re-reduction of P700$^+$ after the initial peak in aerobic conditions (*Figure 4B*) is due to electron donation from PSII in *E. timida* and *Acetabularia*, which further validates the method. The final oxidation phase was absent in both species in anaerobic conditions (*Figure 4D*), indicating that both *E. timida* and *Acetabularia* utilize oxygen-dependent alternative electron sinks functioning after PSI to maintain P700 oxidized.

During the optimization process of the P700 oxidation measurements, we found that firing a second high-light pulse 10 s after the first pulse resulted in a higher capacity to maintain P700 oxidized in both *E. timida* and *Acetabularia* in aerobic conditions (*Figure 4B*, inset). This procedure will hereafter be referred to as 'second pulse protocol'. In *E. timida* the oxidation capacity was moderate even with the second pulse, showing a high re-reduction of P700$^+$ after the initial oxidation. In *Acetabularia*, the second pulse rescued P700 oxidation capacity completely. When a second pulse was fired in anaerobic conditions, P700 oxidation capacity showed only weak signs of improvement in *E. timida* and *Acetabularia* (*Figure 4D*, inset).

## P700 redox kinetics in *E. timida* are affected by the acclimation status of its prey

We tested the sensitivity of *E. timida* P700$^+$ measurements by inflicting changes to the P700 oxidation capacity of its prey, and then estimating whether the changes are present in the slugs after feeding them with differently treated *Acetabularia*. First, we grew *Acetabularia* in elevated (1%) $CO_2$ environment, as high $CO_2$ induces downregulation of the main alternative electron sinks of PSI, FLVs, in cyanobacteria and green algae (*Zhang et al., 2012*; *Jokel et al., 2015*; *Santana-Sanchez et al., 2019*). Next, we allowed *E. timida* to feed on high-$CO_2$ *Acetabularia* for 4 days. We used *E. timida* individuals that had been pre-starved for 4 weeks to ensure that the slugs would only contain chloroplasts from high-$CO_2$ *Acetabularia*. After feeding, the slugs were allowed to incorporate the chloroplasts into their own cells for an overnight dark period in the absence of *Acetabularia* prior to the measurements. A similar treatment was applied to slug individuals that were fed ambient-air grown *Acetabularia* (see 'Materials and methods' for the differences in the feeding regimes). These slugs will hereafter be termed as high-$CO_2$ and ambient-air *E. timida*, respectively.

Ambient-air *E. timida* exhibited stronger P700 oxidation during the initial dark-to-light transition than high-$CO_2$ *E. timida* (*Figure 5A*). When the second pulse protocol was applied, both groups showed a clear increase in P700 oxidation capacity, but once again P700 oxidation was stronger in the ambient-air slugs (*Figure 5C*). The differences between ambient-air and high-$CO_2$ *Acetabularia* showed the same trend (*Figure 5B,D*). Acclimation to high $CO_2$ also caused changes to PSII activity, estimated as relative electron transfer of PSII (rETR) during rapid light curve (RLC) measurements from dark acclimated samples (*Figure 5E,F*). Maximal rETR was lower in ambient-air *E. timida* and *Acetabularia* than in their high-$CO_2$ counterparts. Also the behavior of NPQ during the RLC measurements indicated that ambient-air and high-$CO_2$ *E. timida* had very similar photosynthetic

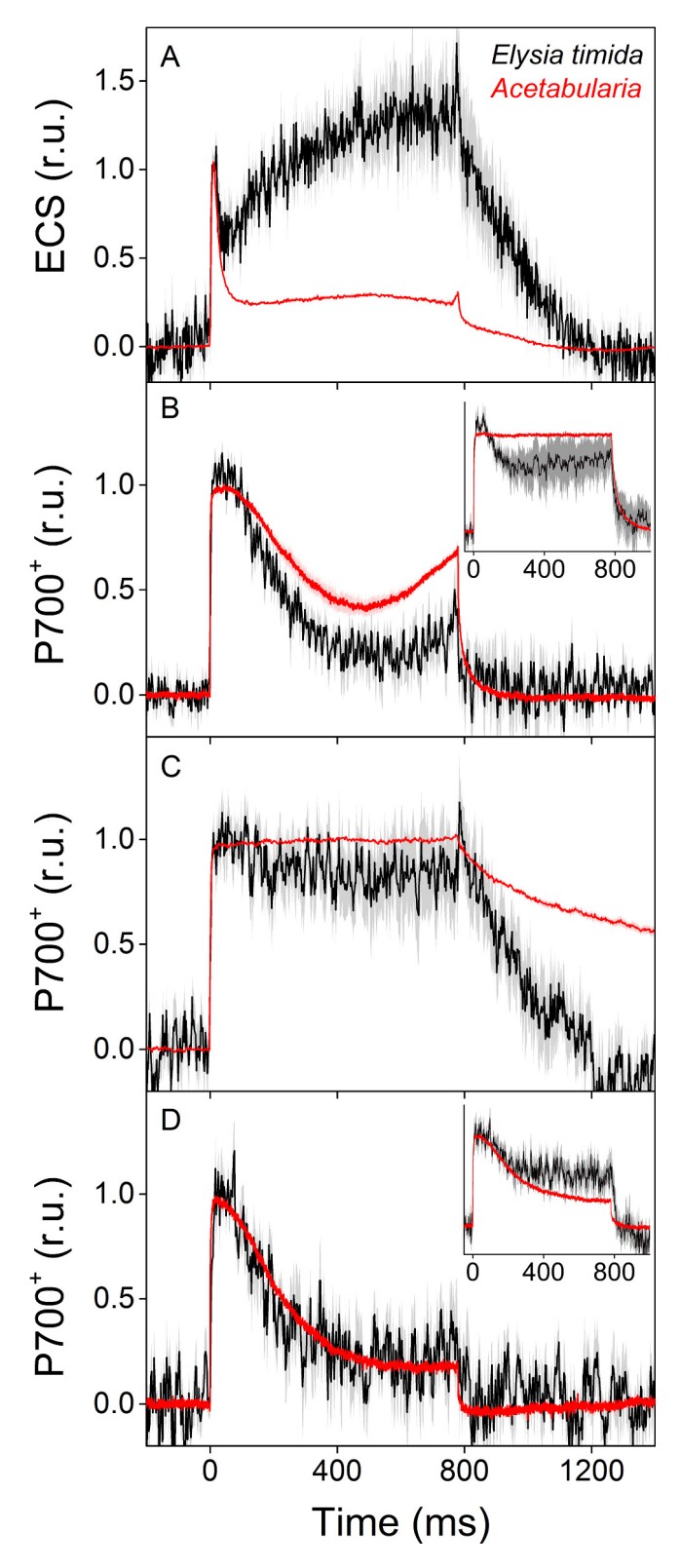

**Figure 4.** ECS and P700$^+$ measurements indicate differences in proton motive force formation and utilization of alternative electron acceptors of PSI between dark acclimated *E. timida* (black) and *Acetabularia* (red) during a 780 ms high-light pulse. (**A**) ECS measured from *E. timida* and *Acetabularia* upon exposure to a high-light pulse in aerobic conditions. (**B**) P700 redox kinetics upon exposure to a high-light pulse in aerobic conditions without any

*Figure 4 continued*

inhibitors. The inset shows P700 redox kinetics from the same samples during a second high-light pulse, fired 10 s after the first one. (C) P700 oxidation kinetics in the presence of 10 μM DCMU. (D) P700 oxidation kinetics in the absence of DCMU in anaerobic conditions, achieved by a combination of glucose oxidase (8 units/ml), glucose (6 mM) and catalase (800 units/ml). The inset shows P700 oxidation kinetics from the same samples during the second high-light pulse. ECS and P700$^+$ transients were double normalized to their respective dark levels (measured prior to the onset of the high-light pulse), and to the initial ECS or P700$^+$ peak (measured immediately after the onset of the pulse). Curves in (A) are averages from 13 (*E. timida*) and 6 (*Acetabularia*) biological replicates, 7 and 3 in (B), 13 and 4 in (C), and 8 (7 in inset) and three in (D), respectively. The shaded areas around the curves represent SE. All *E. timida* data are from individuals taken straight from the feeding tanks, without an overnight starvation period. See *Figure 4—source data 1* for original data.

The online version of this article includes the following source data and figure supplement(s) for figure 4:

**Source data 1.** Source data of ECS and P700+ measurements shown in *Figure 4*.
**Figure supplement 1.** ECS and P700$^+$ signals from bleached *E. timida* individuals during a 780 ms high light pulse.
**Figure supplement 1—source data 1.** Source data of ECS and P700+ measurements shown in *Figure 4—figure supplement 1*.

responses as their respective food sources (*Figure 5G,H*). However, the slugs did exhibit higher levels of NPQ than the algae in both cases.

As regulation of FLVs in response to $CO_2$ conditions might explain the differences in P700 redox kinetics (*Zhang et al., 2012*; *Jokel et al., 2015*; *Santana-Sanchez et al., 2019*), we confirmed their presence in *Acetabularia* by Western blotting (*Figure 6*), using an antibody raised against *Chlamydomonas reinhardtii* FLVB; the antibody also reacts with FLVA of *C. reinhardtii* (*Jokel et al., 2015*). This antibody recognized one protein band of approximately 60 kDa size in *Acetabularia*, falling to the size range of FLVs from *C. reinhardtii* (70 and 58 kDa for FLVA and FLVB, respectively; *Figure 6B*). Our results show that there was no significant difference in FLV amounts between ambient air and high-$CO_2$ *Acetabularia* (*Figure 6A*; Student's t-test, p=0.594, n = 3), suggesting that $CO_2$ concentration does not regulate FLVs at the protein level in *Acetabularia*. However, the FLV-specific band of *Acetabularia* was wider than the Coomassie-stained band (*Figure 6C*), possibly indicating that the antibody reacted with multiple proteins of similar molecular weight. In *C. reinhardtii*, the Coomassie stain and western blot produced a similar FLVB band, but Coomassie staining did not reveal the FLVA band (*Figure 6C*).

## High-$CO_2$ *E. timida* kleptoplasts are sensitive to fluctuating light

Recently, there has been an increasing interest in protection of PSI by alternative electron sinks such as FLVs (*Shimakawa et al., 2019*; *Gerotto et al., 2016*; *Jokel et al., 2018*). This led us to investigate whether P700 oxidation capacity would affect the longevity of the kleptoplasts. We first compared kleptoplast longevity in ambient-air and high-$CO_2$ *E. timida* in starvation under normal day/night cycle (12/12 hr, PPFD 40 μmol m$^{-2}$s$^{-1}$ during daylight hours), that is steady-light conditions. The slugs used here were subjected to a 4-week pre-starvation protocol prior to feeding them with their respective algae before the onset of the actual steady-light starvation experiment (see 'P700 redox kinetics in *E. timida* are affected by the acclimation status of its prey' and 'Materials and methods' for details).

Both groups behaved very similarly in terms of slug coloration and size, maximum quantum yield of PSII photochemistry ($F_V/F_M$) and minimal and maximal chlorophyll fluorescence ($F_0$ and $F_M$, respectively) during a 46-day starvation period (*Figure 7*). Both $F_0$ and $F_M$ decreased during starvation (*Figure 7D*) but as several factors, including photoinhibition of PSII (*Tyystjärvi, 2013*), quenching of excitation energy by photoinhibited PSII (*Matsubara and Chow, 2004*) and the overall decrease of kleptoplasts during starvation affect these values, all conclusions will be based on the $F_V/F_M$ ratio. In both groups, $F_V/F_M$ decreased during starvation in a bi-phasic pattern, with slow decrease until day 21, after which PSII activity declined rapidly. The overall decline in $F_V/F_M$ was nearly identical in both groups throughout the experiment (*Figure 7C*). The initial population size of both groups was 50 slugs, and starvation induced deaths of 9 and 12 slugs during the experiment from ambient-air and high-$CO_2$ *E. timida* populations, respectively (see 'Materials and methods' for

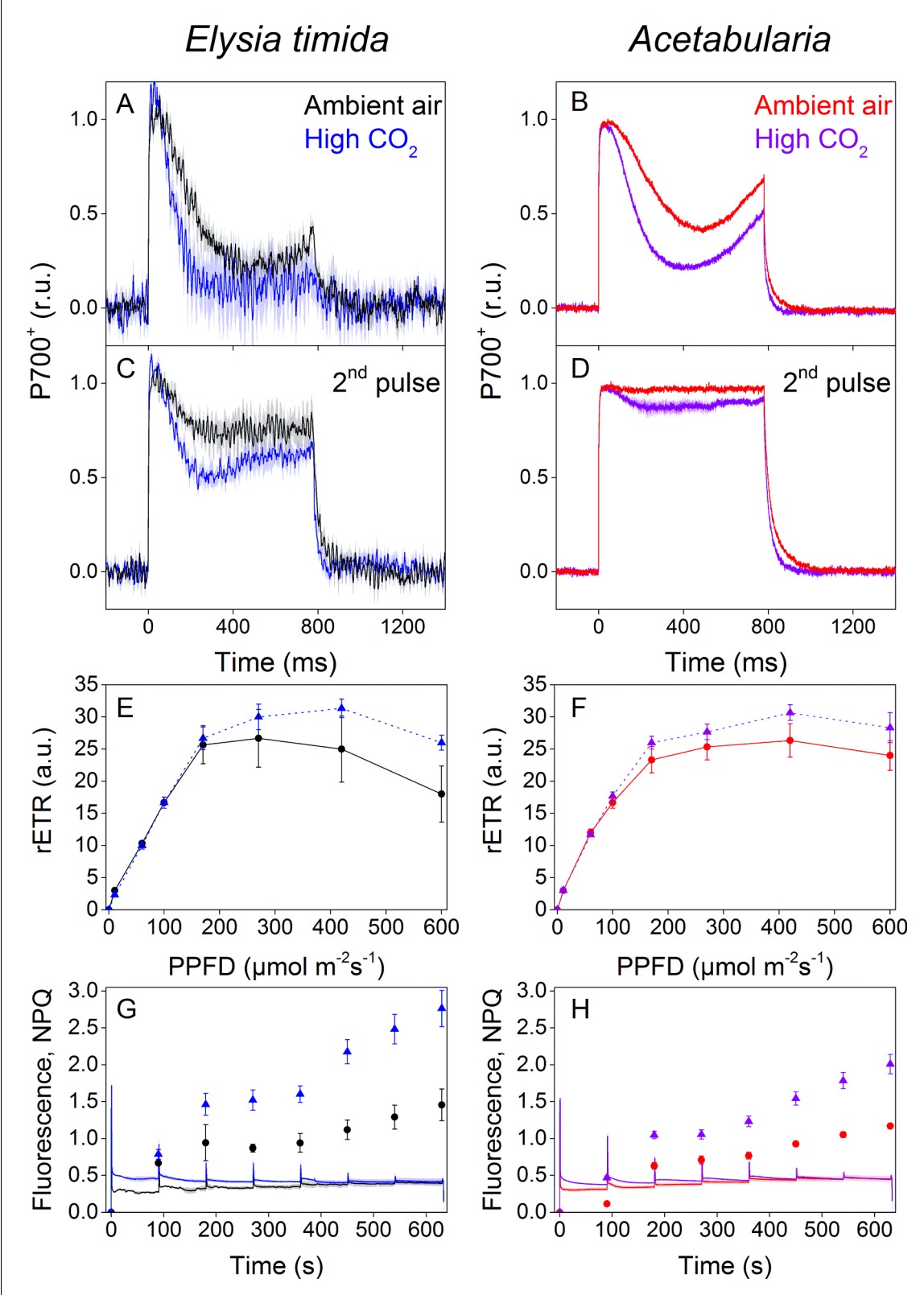

**Figure 5.** P700 redox kinetics, photosynthetic electron transfer and photoprotective NPQ levels in *E. timida* kleptoplasts (left panels) are affected by the $CO_2$ acclimation state of its feedstock *Acetabularia* (right panels). (A–B) P700 redox kinetics in dark acclimated ambient-air (black) and high-$CO_2$ *E. timida* (blue) (A) and ambient-air (red) and high-$CO_2$ *Acetabularia* (purple) (B) upon exposure to a 780 ms high-light pulse. The ambient-air *Acetabularia* data are the same as in *Figure 4B* and are shown here for reference. (C–D) P700 redox kinetics during a second light pulse, fired 10 s after the first

*Figure 5 continued on next page*

*Figure 5 continued*

pulse, shown in panels A-B, in ambient-air and high-$CO_2$ *E. timida* (C) and ambient-air and high-$CO_2$ *Acetabularia* (D). The ambient-air *Acetabularia* data are the same as in *Figure 4B* inset and are shown here for reference. (E–F) RLC measurements from dark acclimated ambient-air (black solid line) and high-$CO_2$ *E. timida* (blue dashed line) (E) and ambient-air (red solid line) and high-$CO_2$ *Acetabularia* (purple dashed line) (F). Illumination at each light intensity (PPFD) was continued for 90 s prior to firing a saturating pulse to determine relative electron transfer rate of PSII (rETR). *E. timida* individuals used in RLC measurements were fixed in 1% alginate for the measurements (see 'Materials and methods' and *Figure 5—figure supplement 1* for details). (G–H) Fluorescence traces and NPQ during the RLC measurements from ambient-air and high-$CO_2$ *E. timida* (G) and ambient-air and high-$CO_2$ *Acetabularia* (H). P700$^+$ transients were double normalized to their respective dark levels and to the P700$^+$ peak measured immediately after the onset of the pulse. Curves in (A) are averages from 7 (ambient-air *E. timida*) and 8 (high-$CO_2$ *E. timida*) biological replicates, 7 and 8 in (C), and 3 and 3 in (E,G), respectively. High-$CO_2$ *Acetabularia* curves in (B,D) are averages from three biological replicates. Ambient-air and high-$CO_2$ *Acetabularia* curves in (F,H) are averages from three biological replicates. Shaded areas around the curves and error bars show SE. rETR and NPQ were calculated as described in 'Materials and methods'. All *E. timida* individuals used in panels (A,C,E,G) were allowed to incorporate the chloroplasts for an overnight dark period in the absence of *Acetabularia* prior to the measurements. See *Figure 5—source data 1* for original data.

The online version of this article includes the following source data and figure supplement(s) for figure 5:

**Source data 1.** Source data of P700+ and fluorescence measurements shown in *Figure 5*.
**Figure supplement 1.** The effect of alginate fixation on the maximum quantum yield of PSII ($F_V/F_M$).

details on mortality and sampling). P700$^+$ measurements from slugs starved for 5 days indicated that the starved ambient-air *E. timida* retained a higher P700 oxidation capacity through starvation than high-$CO_2$ *E. timida*, when the second pulse P700$^+$ kinetics protocol was applied (*Figure 7E*). These results show that altered P700 oxidation capacity does not affect chloroplast longevity in *E. timida* in steady-light conditions.

We repeated the starvation experiment with new populations of ambient-air and high-$CO_2$ *E. timida*, but this time the moderate background illumination (PPFD 40 µmol m$^{-2}$s$^{-1}$) was supplemented every 10 min with a 10 s high-light pulse (PPFD 1500 µmol m$^{-2}$s$^{-1}$) during daylight hours, that is fluctuating light. Slugs used in this experiment were not subjected to a pre-starvation protocol prior to feeding them with their respective algae but were simply allowed to replace their old chloroplasts with new specific ones during 6 days of feeding. The starting population size was 45 slugs for both groups and there were no starvation-caused losses during the whole experiment (see 'Materials and methods' for details on sampling).

Starvation in fluctuating light induced faster onset of the rapid phase of $F_V/F_M$ decrease in both groups (*Figure 8A*) when compared to the steady light starvation experiment (*Figure 7C*). The exact onset of the rapid decline of $F_V/F_M$ was difficult to distinguish, but in ambient-air *E. timida* $F_V/F_M$ decrease accelerated only after ~ 20 days, whereas in high-$CO_2$ *E. timida* the turning point was during days 10–14 (*Figure 8A*). The overall longevity of PSII photochemistry in both groups was, however, very similar, as there was a sudden drop in $F_V/F_M$ in the ambient-air slugs on day 31. $F_0$ behaved identically in the two groups, apart from days 0–4, when $F_0$ of the high-$CO_2$ *E. timida* dropped to the level of the ambient-air *E. timida* $F_0$. At day 10, $F_M$ of the high-$CO_2$ slugs dropped drastically and the groups started to differ. Using the second pulse P700$^+$ measurement protocol, we confirmed that the differences in P700 oxidation capacity were noticeable on days 0 and 6 in starvation also in the populations of slugs used for the fluctuating-light experiment (*Figure 8C,D*). RLC measurements were performed on the slugs on day 10 to inspect the underlying causes of the suddenly decreasing fluorescence parameters $F_V/F_M$ and $F_M$ of the high-$CO_2$ slugs (*Figure 8E*). The situation after 10 days in fluctuating light seemed almost the opposite to the situation on day 0 (*Figure 5E*), that is now the ambient-air slugs showed higher $rETR_{MAX}$ than the high-$CO_2$ slugs (*Figure 8E*). However, the behavior of NPQ during the RLC measurements showed that high-$CO_2$ *E. timida* were still able to generate and maintain stronger NPQ than ambient-air *E. timida* (*Figure 8F*, *Figure 8—figure supplement 1*), although the differences were not as strong as on day 0 (*Figure 5G*). These data indicate that the initial chloroplast acclimation status is retained during starvation, and the decrease in $rETR_{MAX}$ in high-$CO_2$ *E. timida* is likely due to light-induced damage to the photosynthetic apparatus.

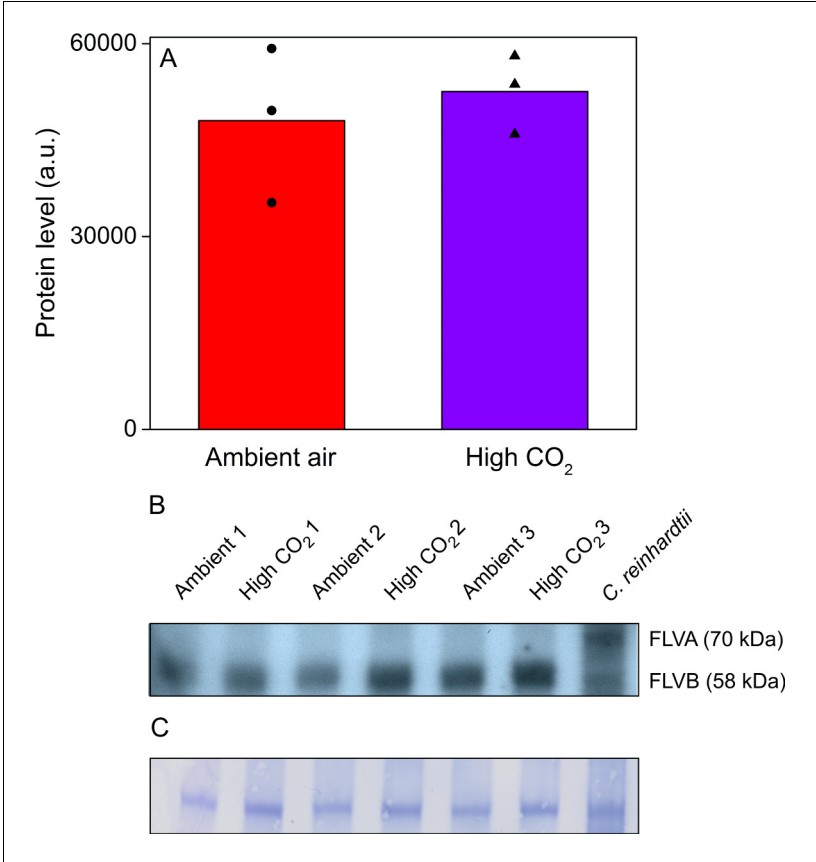

**Figure 6.** *Acetabularia* FLV levels in ambient air and high $CO_2$. (**A**) Densitometric quantification of immunodetected FLV bands from *Acetabularia* grown in ambient air and high $CO_2$ conditions. Bars indicate average protein levels and symbols show original data points. Student's t-test indicated no significant differences between the treatments (p=0.594, n = 3). (**B**) Western blot used for protein quantification in (**A**). FLV proteins were detected from total protein extracts using an antibody raised against *C. reinhardtii* FLVB. The antibody reacts with both FLVA and FLVB proteins of *C. reinhardtii* (*Jokel et al., 2015*). Wild-type *C. reinhardtii* (strain CC406) total protein extract was used as a positive control and FLVA/B bands with their respective sizes are indicated. The lanes in (**B**) represent separate biological replicates. Samples were loaded on total protein basis (25 µg protein/ well). (**C**) Coomassie-stained membrane from the blot in (**B**). See *Figure 6—source data 1* for original data of (**B**). The online version of this article includes the following source data for figure 6:

**Source data 1.** Original blots and membranes of *Figure 6B*.

## Discussion

### Weakened dark reduction of the PQ pool lowers electron pressure in *E. timida* kleptoplasts

Our data on Chl fluorescence relaxation (*Figure 2*) and induction kinetics (*Figure 3*) reveal differences in the dark and light reduction of the photosynthetic electron transfer chain between chloroplasts inside *E. timida* and *Acetabularia*. While the connection between fluorescence transients and specific redox components of the electron transfer chain are still under debate, especially concerning the J-I-P phase of the OJIP curves (*Stirbet and Govindjee, 2012*; *Schansker et al., 2014*; *Vredenberg, 2015*; *Havurinne et al., 2019*; *Magyar et al., 2018*; *Schreiber et al., 2019*), our data suggest that the differences in the redox behavior between the slugs and the algae specifically reflect differences in the redox state of the PQ pool.

The strongest indicators of differences in dark reduction of the PQ pool between the slugs and the algae are the $Q_A^-$ reoxidation measurements from the red morphotype *E. timida* and *Acetabularia* (*Figure 2D*). Red morphotype *Acetabularia* exhibited clear wave like kinetics of fluorescence

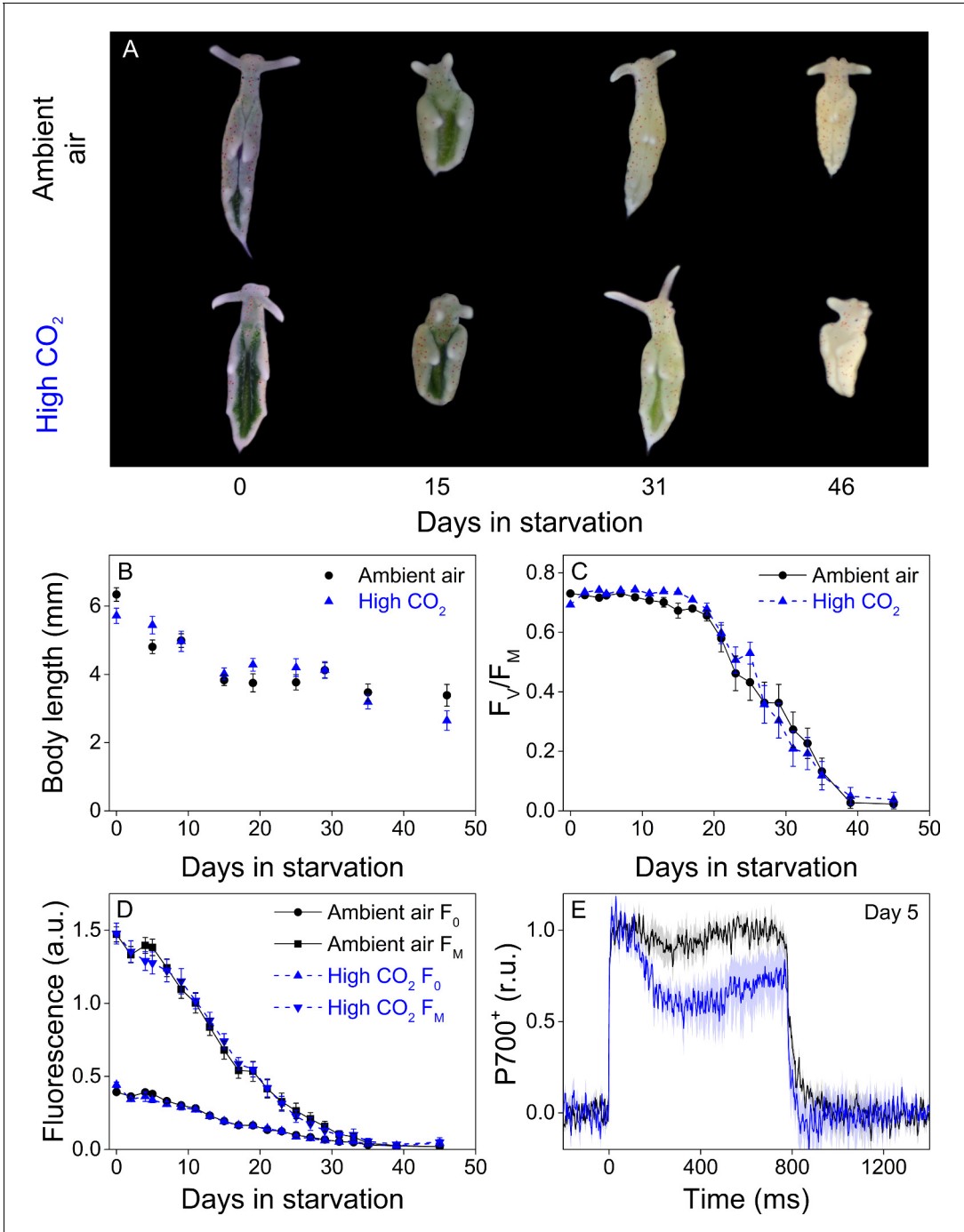

**Figure 7.** Altered P700 oxidation capacity does not affect chloroplast longevity in *E. timida* during starvation in steady-light conditions. (**A–B**) Coloration of selected individuals (**A**) and body length (**B**) of the ambient-air (black) and high-$CO_2$ *E. timida* (blue) slugs during steady-light starvation. The slug individuals in (**A**) do not show the actual scale of the slugs with respect to each other. (**C–D**) Maximum quantum yield of PSII photochemistry ($F_V/F_M$) (**C**) and minimum ($F_0$) and maximum chlorophyll *a* fluorescence ($F_M$) (**D**) during starvation in ambient-air (black) and high-$CO_2$ *E. timida* (blue). (**E**) Second pulse P700 oxidation kinetics after 5 days in steady-light starvation in ambient-air (black) and high-$CO_2$ *E. timida* (blue). Steady-light starvation light regime was 12/12 h day/night and PPFD was 40 μmol $m^{-2}s^{-1}$ during daylight hours. See *Figure 7—figure supplement 1* for the spectra of lamps used in starvation experiments. All data in (**B–D**) represent averages from 50 to 8 biological replicates (see 'Materials and methods' for details on mortality and sampling) and error bars show SE. P700+ transients in were double normalized to their respective dark levels and to the P700+ peak measured immediately after the onset of the pulse, and the curves in (**E**) represent averages from 7 (ambient air *E. timida*) and five biological replicates (high-$CO_2$ *E. timida*) and the shaded areas around the curves show SE. See *Figure 7—source data 1* for original data from panels B-E.

The online version of this article includes the following source data and figure supplement(s) for figure 7:

*Figure 7 continued*
**Source data 1.** Original data from starvation in steady light.
**Figure supplement 1.** Normalized irradiance spectra from different light sources used in the study.

decay, while the wave was completely absent in red *E. timida*. The wave phenomenon was recently characterized in the green alga *C. reinhardtii* (*Krishna et al., 2019*). The authors suggested that, similar to cyanobacteria, non-photochemical reduction of the PQ pool by stromal reductants in anaerobic conditions in the dark leads to wave like kinetics of fluorescence decay after a single turn-over flash in sulphur deprived *C. reinhardtii* cells. The exact mechanisms underlying the wave phenomenon are still unknown, but the involvement of NDA2, a type II NDH protein is clear (*Deák et al., 2014*; *Krishna et al., 2019*). Chemical inhibition of NDA2 in *C. reinhardtii* led to complete abolishment of the wave phenomenon in cells that were otherwise primed for it (*Krishna et al., 2019*). Interestingly, the comparison of NDA2 uninhibited and inhibited cells resulted in fluorescence decay kinetics that are highly reminiscent of the kinetics in *Figure 2D*, with the red morphotype *E. timida* and red morphotype *Acetabularia* being analogous to the NDA2 inhibited

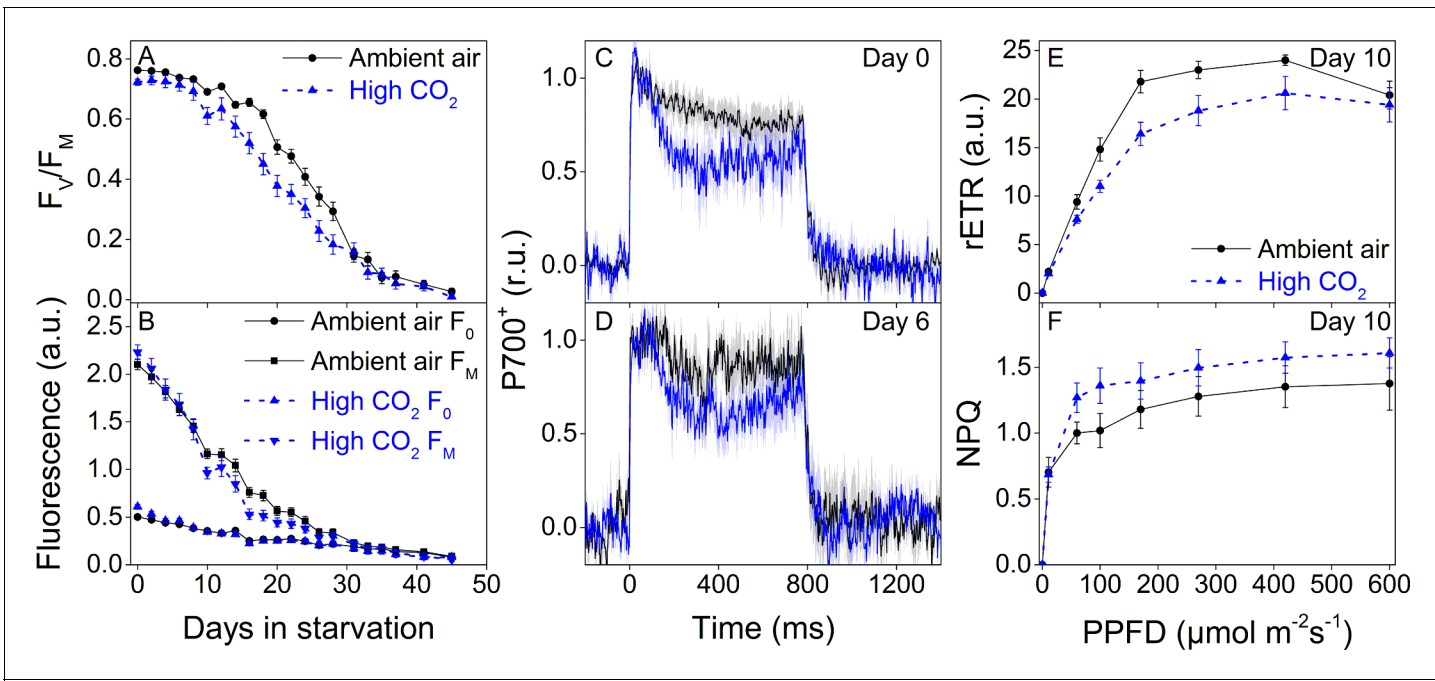

**Figure 8.** Higher P700 oxidation capacity protects the photosynthetic apparatus of ambient-air *E. timida* during fluctuating-light starvation. (A–B) Maximum quantum yield of PSII photochemistry ($F_V/F_M$) (A) and minimum ($F_0$) and maximum chlorophyll fluorescence ($F_M$) (B) during fluctuating light starvation in ambient-air (black solid lines) and high-$CO_2$ *E. timida* (blue dashed lines). (C–D) Second pulse P700 oxidation kinetics after 0 and 5 days in fluctuating-light starvation in ambient-air (black) and high-$CO_2$ *E. timida* (blue). (E–F) Relative electron transfer rate of PSII (rETR) (E) and NPQ (F) during RLC measurement from dark-acclimated ambient-air (black solid lines) and high-$CO_2$ *E. timida* (blue dashed lines) after 10 days in fluctuating-light starvation. Illumination for each light step during the RLCs was continued for 90 s prior to firing a saturating pulse to estimate rETR and NPQ. See *Figure 8—figure supplement 1* for the fluorescence trace and behavior of NPQ during the RLC measurement. The light regime during the fluctuating light starvation was 12/12 hr day/night, and PPFD of the background illumination was 40 μmol m$^{-2}$s$^{-1}$, which was supplemented every 10 min with a 10 s high-light pulse during daylight hours. All data in (A,B) represent averages from 45 to 20 slug individuals (see 'Materials and methods' for details on sampling). P700 redox kinetics in (C) represent averages from nine biological replicates for both ambient-air and high-$CO_2$ *E. timida*, and 6 and 9 in (D), respectively. P700$^+$ transients were double normalized to their respective dark levels and to the P700$^+$ peak measured immediately after the onset of the pulse. Fluorescence-based data in (E,F) represent averages of five biological replicates for ambient-air and high-$CO_2$ *E. timida*. All error bars and shaded areas around the curves show SE. See *Figure 8—source data 1* for original data.

The online version of this article includes the following source data and figure supplement(s) for figure 8:

**Source data 1.** Original data from starvation in fluctuating light.
**Figure supplement 1.** Fluorescence traces and NPQ during the RLC measurement.

and uninhibited *C. reinhardtii* cells, respectively. Despite the remarkable analogy, more experiments are needed to validate the presence of a type II NDH protein in *Acetabularia* and its role in the wave-phenomenon in the red morphotype. However, since there are no reports indicating incorporation of algal mitochondria by Sacoglossan sea slugs and they have been shown to digest cellular components of the algae other than the chloroplasts (*Rumpho et al., 2000*), we interpret the lack of the wave phenomenon as a sign of missing contribution of respiratory electron donors into the kleptoplasts inside *E. timida*. Either *E. timida* kleptoplasts are completely cut off from respiratory electron donors deriving from the slug's own mitochondria, or they are not delivered into the PQ pool due to inhibition of a type II NDH protein. The $Q_A^-$ reoxidation data in *Figure 2C*, showing that fluorescence decay is faster in *E. timida* than in *Acetabularia* in anaerobic conditions, indicate that dark reduction of the PQ pool is weakened also in regular green *E. timida* individuals.

The weakened dark reduction of the PQ pool in *E. timida* has consequences to photosynthetic electron transfer in the light. The ~ 300 ms delay in reaching the maximum fluorescence during the OJIP measurements in *E. timida* in *Figure 3A* indicates that the PQ pool is more oxidized in the slugs than in *Acetabularia* even in aerobic conditions, and full reduction of the PQ pool simply takes longer in *E. timida*. We base this on the notion that full reduction of the PQ pool is a prerequisite for reaching maximum fluorescence when fluorescence rise kinetics are measured using multiple turn-over saturating light pulses, such as the ones used in the current study (*Kramer et al., 1995*; *Yaakoubd et al., 2002*; *Suggett et al., 2003*; *Osmond et al., 2017*). Maintaining an oxidized PQ pool could be advantageous for chloroplast longevity, as it would help mitigate electron pressure and ROS formation in PSII.

## Build-up of proton-motive force in *E. timida* may facilitate strong NPQ

We witnessed a clear increase in proton-motive force during a high light pulse in *E. timida*, whereas in *Acetabularia* proton-motive force relaxes considerably after the initial spike in thylakoid membrane energization (*Figure 4A*). The build-up of proton-motive force in *E. timida* suggests that protons are released into the thylakoid lumen during illumination, but not out. This could be indicative of defects in ATP-synthase functionality in the slugs, perhaps due to lack of inorganic phosphate. Interestingly, our results from ambient air and high-$CO_2$ slugs and algae show that while the acclimation state of the prey algae is reflected onto the NPQ capacity of the slugs, the slugs exhibit higher levels of NPQ than their respective prey *Acetabularia* at the same light intensities (*Figure 5G,H*). Similar data has previously been presented from *E. timida* and also *E. chlorotica* (*Cruz et al., 2015*; *Christa et al., 2018*). Higher level of NPQ in the slugs is likely linked to the strong acidification of the lumen in the slugs (*Figure 4A*), since the major qE component of NPQ is pH dependent (*Müller et al., 2001*; *Papageorgiou GC, 2014*). The xanthophyll cycle of freshly fed *E. timida* is functional (*Cartaxana et al., 2019*), but it is unclear whether the NPQ induced already during the first 100 s of the RLC measurement (*Figure 5G*) is due to lumen acidification switching on the xanthophyll cycle, or whether lumen acidification directly enhances quenching capacity at the level of light harvesting antennae. Nevertheless, strong NPQ has obvious benefits for chloroplast longevity in *E. timida*.

## Flavodiiron proteins function as alternative electron sinks in *E. timida* and *Acetabularia*

*E. timida* and *Acetabularia* utilize oxygen-dependent electron acceptors of PSI during dark-to-light transition, but their functionality is weaker in *E. timida* (*Figure 4B–C*). Based on the current literature on P700 redox kinetics during the first seconds after a dark to light transition, the most likely candidates for these electron acceptors are FLVs that donate electrons to oxygen (*Jokel et al., 2015*; *Gerotto et al., 2016*; *Ilík et al., 2017*; *Jokel et al., 2018*; *Shimakawa et al., 2019*). Indeed, the presence of FLVs was also confirmed in *Acetabularia* by western blotting (*Figure 6*). Using the second pulse protocol for P700 redox measurements, we showed that P700 oxidation capacity increases after the initial pulse of light in both *E. timida* and *Acetabularia* (*Figure 4B*). Full activation of FLVs as electron acceptors of PSI takes ~ 1 s after a transition from dark to light and they subsequently remain a considerable electron sink during the time required for light activation of Calvin-Benson-Bassham cycle (*Ilík et al., 2017*; *Gerotto et al., 2016*; *Bulychev et al., 2018*). The mechanism of such fast regulation of FLV functionality is not known, but the conserved cysteine residues of

FLVs have been suggested to offer a means for redox regulation through conformational changes (*Alboresi et al., 2019*). This, in conjunction with the data showing that anaerobicity diminishes P700 reoxidation even with the second pulse protocol (*Figure 4D*), support the view that FLVs are also behind P700 oxidation during the second pulse.

## High P700 oxidation capacity improves kleptoplast longevity under fluctuating light

By feeding *E. timida* individuals with *Acetabularia* grown in ambient air and high $CO_2$ conditions, we successfully created slugs whose kleptoplasts reflected the acclimation state of their respective prey algae at the levels of P700 redox kinetics (*Figure 5A–D*), rETR (*Figure 5E,F*) and NPQ (*Figure 5G, D*). Acclimation to high $CO_2$ led to a noticeable decrease in P700 oxidation capacity in both *Acetabularia* and *E. timida* (*Figure 5A–D*). The simplest explanation for this is downregulation of FLVs in elevated $CO_2$ environment (*Jokel et al., 2015*), but this was not confirmed by a western blot analysis from ambient-air and high-$CO_2$ *Acetabularia* (*Figure 6A*). However, the protein band detected with the FLV antibody from *Acetabularia* is broad, and might conceal two protein bands, possibly the *Acetabularia* homologues of FLVA and FLVB. Another important factor influencing PSI electron transfer that could be affected by alterations in $CO_2$ levels in *Acetabularia* is the Mehler's reaction, but this seems an unlikely candidate for the differences in P700 redox kinetics during the 780 ms light pulse, as the evidence from FLV knockout and insertion mutant algae and plants show that Mehler's reaction cannot substitute for FLVs in the reoxidation of P700 during such a time frame (*Gerotto et al., 2016*; *Ilík et al., 2017*; *Jokel et al., 2018*). Also altered stoichiometry of PSII and PSI could affect P700 redox kinetics, but the highly similar ratio of chlorophylls *a* and *b* between ambient-air (chlorophyll *a/b* = 2.37, SE ± 0.03, n = 5) and high-$CO_2$ slugs (chlorophyll *a/b* = 2.41, SE ± 0.06, n = 6) indicates that photosystem stoichiometry does not strongly depend on $CO_2$ concentration. Whatever the exact reason behind the altered P700 redox kinetics is, it is clear that acclimation of *Acetabularia* to high $CO_2$ lowers P700 oxidation capacity and this acclimation state is transferred into *E. timida*.

When ambient air and high-$CO_2$ *E, timida* were starved for 46 days in steady-light conditions, they behaved nearly identically in terms of kleptoplast longevity, while still retaining the differences in P700 redox kinetics after 5 days in starvation (*Figure 7*). The differences in P700 redox kinetics were also noticeable on days 0 and 6 of the starvation experiment in fluctuating light. In fluctuating light, $F_V/F_M$ of the high-$CO_2$ slugs decreased faster than in ambient-air slugs (*Figure 8*). Our results suggest that alternative electron acceptors of PSI, possibly FLVs, are utilized by *E. timida* to protect kleptoplasts from formation of ROS during fluctuating-light starvation. The exact mechanism of PSI damage is not clear, but FLVs have been shown to protect PSI in green algae by donating excess electrons to oxygen without producing ROS (*Shimakawa et al., 2019*; *Jokel et al., 2018*). However, both slug groups did retain PSI activity for at least up to 6 days in starvation (*Figure 8D*). This shows that PSI was protected against fluctuating light even in the high-$CO_2$ *E. timida* that exhibited lowered, but not completely abolished, P700 oxidation capacity. Lower P700 oxidation capacity in high-$CO_2$ slugs could, however, cause an increase in the rate of Mehler's reaction (*Mehler, 1951*; *Khorobrykh et al., 2020*). Superoxide anion radical and hydrogen peroxide, the main ROS produced by Mehler's reaction, are not likely to be involved in the primary reactions of PSII photoinhibition but are known to have deleterious effects on PSII repair (*Tyystjärvi, 2013*). We propose that this is behind the faster decrease in $F_V/F_M$ and rETR in the high-$CO_2$ *E. timida* in fluctuating light.

## Conclusions

We have performed the most detailed analysis of the differences in photosynthetic light reactions between a photosynthetic sea slug and its prey alga to date. Our results indicate that in the dark the PQ pool of the kleptoplasts inside the sea slug *E. timida* is not reduced to the same extent as in chloroplasts inside the green alga *Acetabularia* (*Figure 2*). Fluorescence induction measurements also suggest that there are differences in the PQ pool redox state between kleptoplasts in *E. timida* and chloroplasts in *Acetabularia*. The considerable delay in reaching the maximum chlorophyll *a* fluorescence in *E. timida* during a high-light pulse (*Figure 3A*) can be indicative of a highly oxidized PQ pool that simply takes longer to fully reduce.

Our results show that oxidation of P700 seems to be weaker in *E. timida* than in *Acetabularia* (*Figure 4B*, *Figure 5A–D*), but still enough to offer protection from light-induced damage in fluctuating light in *E. timida* (*Figure 8*). If the capacity to oxidize P700 by alternative electron acceptors is lowered in *E. timida* kleptoplasts, is this compensated for by an increased capacity of the main electron sink, that is the Calvin-Benson-Bassham cycle? If not, *E. timida* slugs would risk having a foreign organelle inside their own cells that readily produces ROS via one-electron reduction of oxygen. Interestingly, a major feature separating Sacoglossan slug species capable of long-term retention of kleptoplasts from those that are not, is their high capacity to downplay starvation-induced ROS accumulation (*de Vries et al., 2015*). This could imply that long-term retention slug species such as *E. timida* do not need to concern themselves over the perfect functionality of the electron transfer reactions downstream of PSI. Further in-depth investigations into the carbon fixation reactions in photosynthetic sea slugs are needed to test this hypothesis. In addition to bringing closure to a biological conundrum that has remained unanswered for decades, solving how sea slugs are able to incorporate and maintain kleptoplasts in their own cells could provide useful insights into the ancient endosymbiotic events that led to the evolution of eukaryotic life.

# Materials and methods

## Key resources table

| Reagent type (species) or resource | Designation | Source or reference | Identifiers | Additional information |
|---|---|---|---|---|
| Strain, strain background (*Elysia timida*) | *Elysia timida* Turku Isolate1 (TI1) | This paper | | Mediterranian locality (Elba, Italy, 42.7782° N, 10.1927° E) |
| Strain, strain background (*Acetabularia acetabulum*) | *Acetabularia acetabulum* Düsseldorf Isolate 1 (DI1) | This paper, *Schmitt et al., 2014* | | Mediterranian locality, s train originally isolated by Diedrik Menzel |
| Biological sample (*Chlamydomonas reinhardtii*) | CC406 wild-type strain | *Jokel et al., 2015*; Chlamydomonas resource center | RRID:SCR_014960 | Total protein extract |
| Antibody | Rabbit anti-FLVB | *Jokel et al., 2015* | | (1:5000) |
| Commercial assay or kit | DC Protein Assay Kit | Bio-Rad | Bio-Rad #5000111 | |
| Commercial assay or kit | Next Gel 10% Polyacrylamide Gel Electrophoresis Solutions | VWR | VWR # 97063–026 | |
| Chemical compound, drug | 3-(3, 4-dichlorophenyl)−1, 1-dimethylurea (DCMU) | Merck | Merck #D2425 | |
| Chemical compound, drug | D-(+)-Glucose | Merck | Merck #G8270 | |
| Chemical compound, drug | Glucose Oxidase from *Aspergillus niger* | Merck | Merck #G2133 | |
| Chemical compound, drug | Catalase from bovine liver | Merck | Merck #C9322 | |
| Software, algorithm | Fiji | *Schindelin et al., 2012* | RRID:SCR_003070 | |
| Software, algorithm | Origin | Originlab (https://originlab.com) | RRID:SCR_014212 | Origin 2016 v.9.3 |

## Organisms and culture conditions

Axenic stock cultures of the green alga *Acetabularia* (Düsseldorf Isolate 1, DI1; strain originally isolated by Diedrik Menzel) were grown in 5–10 l plastic tanks in sterile filtered f/2 culture medium made in 3.7% artificial sea water (ASW; Sea Salt Classic, Tropic Marin, Montague, MA, USA). All ASW were prepared in 10 l batches in regular tap water of Turku, Finland. It is important to note the regional differences in tap water quality, and the suitability of tap water for cultures of algae or sea

slugs should be assessed before large scale cultures to avoid mass loss of the cultures. In order to slow down the stock culture growth, PPFD of growth lights (TL-D 58W/840 New Generation fluorescent tube; Philips, Amsterdam, The Netherlands) was < 20 µmol m$^{-2}$s$^{-1}$. The culture medium for the stock cultures was changed at 8–10 week intervals. Keeping the lids of the *Acetabularia* tanks locked in place (*Figure 1E*) effectively prevented excessive evaporation from the tanks. Other *Acetabularia* culture maintenance procedures, such as induction of gamete release, formation of zygotes and sterilization procedures were performed essentially as described earlier (*Hunt and Mandoli, 1992*; *Cooper and Mandoli, 1999*). The day-night cycle was 12/12 hr and temperature was maintained at 23°C at all times for all algae and slug cultures, unless mentioned otherwise. Algae used in the experiments were transferred to new tanks containing fresh f/2 media and grown under lights adjusted to PPFD 40 µmol m$^{-2}$s$^{-1}$ (TL-D 58W/840 New Generation fluorescent tube) for minimum of two weeks prior to any further treatments. No attempt was made to use only algae of certain age or size, and all populations were mixtures of cells in different developmental stages. PPFD was measured with a planar light sensor (LI-190R Quantum Sensor; LI-COR Biosciences; Lincoln, NE, USA) at the tank bottom level in all growth and treatment conditions. Irradiance spectra of all growth light sources used in the current study are shown in *Figure 7—figure supplement 1, measured with an absolutely calibrated STS-VIS spectrometer (Ocean Optics, Largo, FL, USA).*

Sea slug *E. timida* individuals (50 individuals in total) were initially collected from the Mediterranean (Elba, Italy, 42.7782° N, 10.1927° E). The slug cultures were routinely maintained essentially as described by *Schmitt et al., 2014*. Briefly, *E. timida* were maintained at the same conditions as the cultures of *Acetabularia*, their prey alga, in aerated 5–10 l plastic tanks containing 3.7% ASW. Newly prepared ASW was added to the tanks weekly to dilute contaminants and slug excrements, and the slugs were placed in new tanks with fresh ASW at 3–5 week intervals. Similar to *Acetabularia* tanks, also the lids of the slug tanks were locked in place. The tanks were aerated with a silicon tubes connected to air spargers (*Figure 1E*). Differing amounts of *Acetabularia* were added to the slug tanks at irregular intervals, usually once every two weeks. When the adult slugs were transferred to new tanks, the old tanks with their contents were not discarded but supplemented with fresh ASW media and *Acetabularia* in order to allow unhatched slugs or slugs still in their veliger stage to develop into juvenile/adult slugs that are visible to the eye and could be pipetted out with a 10 ml plastic Pasteur pipette. The development from microscopic veligers to juvenile slugs usually took 2–3 weeks. Our method for cultivating slugs has enabled us to maintain a constant slug population consisting of 500–1000 slugs with relatively little labour and cost for years. It is, however, difficult to maintain the slug cultures axenic, and the slug tanks do foul, if not attended to. The contaminants in our laboratory cultures have not yet been identified but, based on optical inspection, seem to consist mainly of diatoms and ciliates. These organisms are likely derived from the Mediterranean and have been co-cultured with the slugs throughout the years. All slugs used in the experiments were always transferred into new tanks filled with fresh ASW and fed with abundant *Acetabularia* for 1–2 weeks prior to the experiments, unless mentioned otherwise. Slug individuals taken straight from the normal culture conditions were used for some of the measurements, without special considerations on the retention status of the chloroplasts inside the slugs, that is the slugs were not allowed to incorporate the chloroplasts overnight in the dark. The use of slugs without overnight settling time is indicated in the figures.

Acclimating *Acetabularia* to elevated $CO_2$ levels was done in a closed culture cabinet (Algaetron AG230; Photon Systems Instruments, Drásov, Czech Republic) by raising the $CO_2$ level from the ambient concentration (0.04% of air volume) to 1% $CO_2$ inside the cabinet. Plastic 5 l tanks filled with *Acetabularia* were placed inside the cabinet and the tank lids were slightly opened, that is one of the plastic lid locks was not locked, to facilitate gas exchange. Newly made f/2 medium was added every second day to account for evaporation (approximately 50 ml per week, estimated in a separate tank) as a precaution. This can admittedly lead to salinity fluctuations, but we chose to add f/2 medium instead of water, as the evaporation in our conditions was low. Incident light provided by the growth cabinet white LEDs (see *Figure 7—figure supplement 1A* for the spectrum) was adjusted to PPFD 40 µmol m$^{-2}$s$^{-1}$ and the day-night cycle was 12/12 hr. Temperature was maintained at 23°C. The algae were always acclimated for a minimum of three days to high $CO_2$ prior to any measurements or feeding of the slugs with high-$CO_2$ acclimated algae.

The red morphotype of *Acetabularia* was induced by growing the algae at 10°C and continuous high white light (PPFD 600 µmol m$^{-2}$s$^{-1}$) for 31 days in a closed culture cabinet in ambient air

(Algaetron AG230; Photon Systems Instruments), essentially as described by *Costa et al., 2012*. While this procedure was successful in producing the desired colour morphotype of *Acetabularia*, the yield was very low, and most of the *Acetabularia* cells bleached during the treatment. A few cells of red *Acetabularia* could be found in tanks where the cell concentration had been high enough to create a light attenuating algal mat. These red *Acetabularia* cells were then collected and used for measurements or fed to the slugs as indicated.

*E. timida* individuals that were devoid of chloroplasts, that is bleached, were obtained by subjecting freshly fed slugs to starvation in high light (Ikea Växer PAR30 E27, 10 W; PPFD > 1000 µmol m$^{-2}$s$^{-1}$; 12/12 hr day/night) in otherwise normal growth conditions. After 1 week of starvation the coloration of the slugs was estimated by eye, and only the palest individuals were selected for further experiments.

## Protein analysis

Crude total proteins were extracted from *Acetabularia* cells grown in ambient air and high $CO_2$ conditions by cutting up approximately 4 g (wet weight) of algae with a razor blade and then grinding them in 500 µl of lysis buffer (50 mM Tris-HCl pH 8, 2% SDS, 10 mM EDTA) using a 1 ml Dounce tissue grinder (DWK Life Sciences, Millville, NJ, USA). The homogenate was filtered through one layer of Miracloth (Calbiochem, Darmstadt, Germany). Algae used for the extractions were taken straight from their respective growth conditions. All protein isolation procedures were performed at 4°C in dim light. Protein concentrations from the homogenates were determined with DC Protein Assay (Bio-Rad, Hercules, CA, USA) using BSA as a standard.

Aliquots containing 25 µg protein were solubilized and separated by electrophoresis on a 10% SDS-polyacrylamide gel using Next Gel solutions and buffers according to the manufacturer's instructions (VWR, Radnor, PA, USA). Proteins were transferred to Immobilon-P PVDF membranes (MilliporeSigma, Burlington, MA, USA). FLVs were immunodetected using an antibody raised against *C. reinhardtii* FLVB, reacting with *C. reinhardtii* FLVA and FLVB (*Jokel et al., 2015*). Western blots were imaged using goat anti-rabbit IgG (H+L) alkaline phosphatase conjugate (Life Technologies, Carlsbad, CA, USA) and CDP-star Chemiluminescence Reagent (Perkin-Elmer, Waltham, MA, USA). Protein bands were quantified using Fiji image processing software (*Schindelin et al., 2012*). Total protein extract from wild-type *C. reinhardtii* strain CC406 was used as a positive control of FLVs (*Jokel et al., 2015*).

## Fast kinetics of Q$_A^-$ reoxidation, fluorescence induction (OJIP), P700 oxidation and ECS

Algae and slug samples were dark acclimated for 1–2 hr prior to the fast kinetics measurements and all fast kinetics were measured at room temperature. PSII electron transfer was blocked in certain measurements, as indicated in the figures, with DCMU. For this, a 2 mM stock solution of DCMU in dimethylsulfoxide was prepared and diluted to a final concentration of 10 µM in either f/2 or ASW medium, depending on whether it was administered to the algae or the slugs, respectively. DCMU was only applied to samples that had been in the dark for 1 hr and the dark acclimation in the presence of DCMU was continued for additional 20 min. When pertinent, DCMU containing medium was applied to cover the samples during the actual measurements too.

Anaerobic conditions were achieved by a combination of glucose oxidase (eight units/ml), glucose (6 mM) and catalase (800 units/ml) in f/2 or ASW medium. Our data shows that using the above reagents and concentrations in a sealed vial with stirring, nearly all oxygen was consumed from 250 ml of ASW media in a matter of minutes (*Figure 2—figure supplement 1A*). Similar to the DCMU treatments, anaerobic conditions were initiated only after the samples had been in the dark for 1 hr. In the case of *Acetabularia*, the samples were placed inside a sealed 50 ml centrifuge tube filled with f/2 medium that had been pre-treated with the glucose oxidase system for 10 min. The algae were then kept inside the sealed tube for 5 min in the dark, after which they were picked out and placed to the sample holder of the instrument in question. In order to maintain oxygen concentrations as low as possible, the samples were then covered with the anaerobic medium and left in the dark for additional 5 min, so that the oxygen mixed into the medium during sample placement would be depleted. Before imposing anaerobic conditions to slug individuals, they were swiftly decapitated with a razor blade, a procedure that has been shown not to significantly affect PSII activity in the

photosynthetic sea slug *Elysia viridis* during a 2 hr measurement period (*Cruz et al., 2015*). The euthanized slugs were then treated identically to the algae used in the anaerobic measurements.

$Q_A^-$ reoxidation kinetics after a strong single turnover (ST) flash (maximum PPFD 100,000 µmol m$^{-2}$s$^{-1}$, according to the manufacturer) were measured using an FL 200 fluorometer with a Super-Head optical unit (Photon Systems Instruments, Drásov, Czech Republic) utilizing the software and protocol provided by the manufacturer. The measurement protocol was optimized to be robust enough to allow its use in measurements from both *Acetabularia* and the slugs. The parameters used in the script were as follows: experiment duration - 120 s, Number of datapoints/decade - 8, First datapoint after ST flash - 150 µs, ST flash voltage – 100%, ST flash duration – 30 µs, measuring beam (MB) voltage – 60%. The wavelength for the ST flash and the MB was 625 nm. The option to enhance the ST flash intensity by complementing it with the MB light source was not used in the measurements. Number of datapoints/decade was changed to two for the measurements in the presence of DCMU.

The slugs tend to crawl around any typical cuboid 2 ml measuring cuvette if the cuvette is filled with ASW, which causes disturbances to the fluorescence signal. On the other hand, if ASW is removed from the cuvette, the slugs tend to stick to the bottom, placing them away from the light path of the instrument. For this reason, a compromise was made between ideal optics and slug immobilization by placing three to five slugs into a regular 1.5 ml microcentrifuge tube and then pipetting most of the ASW out of the tube, leaving just enough ASW to cover the slugs. Only in the measurements in anaerobic conditions were the tubes filled with oxygen depleted ASW. The tube was placed into the cuvette holder of the SuperHead optical unit so that the narrow bottom of the tube with the slugs was situated in the middle of the light path of the instrument and the tube was resting on its top appendices. The tube caps were left open for the measurements without any inhibitors and in the presence of DCMU, unlike the anaerobic measurements where the caps were closed. In the context of $Q_A^-$ reoxidation data from the slugs, one biological replicate refers to one measurement from three to five slugs inside the same tube in this study. Completely new slugs were used for each biological replicate. In order to facilitate comparison, $Q_A^-$ reoxidation from the algae was also measured using 1.5 ml microcentrifuge tubes, but due to the sessile nature of the algae there was no need to remove the f/2 media from the tubes. For each biological replicate representing the algae in the $Q_A^-$ reoxidation data sets, approximately five to ten cells were placed inside each of the tubes.

The polyphasic fluorescence rise kinetics (OJIP curves) were measured with AquaPen-P AP 110 P fluorometer (Photon Systems Instruments) that has an inbuilt LED emitter providing 455 nm light for the measurements. The fluorometer was mounted on a stand and all measurements were done by placing a Petri dish with the sample on it on a matte black surface and positioning the sample directly under the probe head of the fluorometer. The intensity of the 2 s multiple turnover (MT) saturating pulse used for the measurements was optimized separately for measurements from single slug individuals (representing one biological replicate) and one to two cells/strands of algae (representing one biological replicate) placed under the probe of the instrument.

The final MT pulse intensity setting of the instrument was 70% (100% being equal to PPFD 3000 µmol m$^{-2}$s$^{-1}$ according to the manufacturer's specifications) for the slugs and 50% for *Acetabularia*. OJIP curves were measured from samples that had been covered in their respective treatment media, which presented a concern with regard to the anaerobic measurements, as the samples were not in a closed environment and oxygen diffusion into the sample could not be prevented. The data in *Figure 2—figure supplement 1B* shows that diffusion is largely negated during the additional 5-min dark period even in an open setup. The conditions during these OJIP measurements will be referred to as anaerobic although some diffusion of oxygen to the samples occurred.

Fast kinetics of P700 oxidation during a 780 ms MT pulse were measured with Dual-PAM 100 (Heinz Walz GmbH, Effeltrich, Germany) equipped with the linear positioning system stand 3010-DUAL/B designed for plant leaves and DUAL-E measuring head that detects absorbance changes at 830 nm (using 870 nm as a reference wavelength). The absorbance changes are not caused entirely by P700 redox state, as the contribution of other components of the electron transfer chain, plastocyanin and ferredoxin, cannot be distinguished from the P700 signal at this wavelength region (*Klughammer and Schreiber, 2016*). P700 measurements were carried out essentially as described by *Shimakawa et al., 2019*, with slight modifications. We built a custom sample holder frame that can be sealed from the top and bottom with two microscope slide cover glasses by sliding the cover

glasses into the frame. The frame of this sample holder was wide enough so that the soft stoppers of the Dual-PAM detector unit's light guide could rest on it without disturbing the sample even when the top cover glass was not in place. A 3D-printable file for the sample holder is available at https://seafile.utu.fi/d/2bf6b91e85644daeb064/.

The 635 nm light provided by the LED array of Dual-PAM was used for the MT pulse (780 ms, PPFD 10,000 µmol m$^{-2}$s$^{-1}$) in the P700$^+$ measurements. Fluorescence was not measured during the MT pulse, as the MB used for fluorescence seemed to disturb the P700$^+$ signal from the slugs. The drift of the signal made attempts to estimate the maximum oxidation level of P700 according to the standard protocol described by *Schreiber and Klughammer, 2008a* impossible with the slugs. Measurements from the algae would not have required any special considerations due to a stronger signal, but the algae were nevertheless measured identically to the slugs for the sake of comparability. The measuring light intensity used for detecting the P700 absorbance changes had to be adjusted individually for each sample. All P700$^+$ kinetics were measured from individual slugs (i.e one slug represents one biological replicate), as pooling multiple slugs together for a single measurement did not noticeably enhance the signal.

Due to the delicate nature of the P700$^+$ signal, all slugs used for these measurements had to be decapitated with a razor blade before the measurements. It is important to note that obtaining a single, meaningful fast kinetics curve of P700 oxidation requires sacrificing a lot of slug individuals. In this study, a minimum of 10 individuals were used to construct each curve, because in approximately 30–50% of the measurements the signal was simply too noisy and drifting to contain any meaningful information.

The P700$^+$ measurements were carried out similarly to the OJIP measurements, with three main differences. First, the number of algae cells per measurement (representing one biological replicate) was higher, usually five to ten cells/strands forming an almost solid green area between the light guides of Dual-PAM inside the sample holder. Secondly, the anaerobic measurements were carried out in a sealed system, achieved by closing the sample holder with both cover glasses after filling it with anaerobic medium. Measurements from all other treatments were carried out in open sample holders. The third difference was that for some of the experiments a second MT pulse was fired after a 10 s dark period following the first MT pulse. This procedure is referred to as 'second pulse P700 redox kinetics protocol' in the main text.

Electrochromic shift (ECS, or P515) during a MT pulse (780 ms, 635 nm, PPFD 10000 µmol m$^{-2}$s$^{-1}$) was measured with P515 module of Dual-PAM 100 using the dual beam 550–515 transmittance difference signal (actual wavelengths used were 550 and 520 nm) (*Schreiber and Klughammer, 2008b*; *Klughammer et al., 2013*). ECS from *Acetabularia* was measured using the exact same setup as with the P700 measurements, but ECS from the slug *E. timida* could only be measured using the pinhole accessory of Dual-PAM 100. Shortly, a pinhole plug was placed on the optical rod of the P515 detector and 3–5 decapitated slug individuals (representing one biological replicate) were placed into the hole of the plug, covering the optical path. After placing a sample between the optical rods of the P515 module, the ECS signal was calibrated and the MB was turned off to decrease the actinic effect caused by the MB. MB was turned back on again right before measuring the ECS kinetics during a MT pulse. The intensity of the MB was adjusted for each sample separately.

The P700 oxidation and ECS data from *E. timida* and *Acetabularia* were slope corrected, when needed, using the baseline subtraction tool of Origin 2016 v.9.3 (OriginLab Corporation, Northampton, MA, USA) to account for signal drift. All biological replicates used to construct the fast kinetics data figures (Q$_A^-$ reoxidation, OJIP, P700 oxidation and ECS) were normalized individually as indicated in the main text figures, and the normalized data were averaged to facilitate comparison between the samples.

## Maximum quantum yield of PSII and rapid light response curves

Maximum quantum yield of PSII photochemistry (F$_V$/F$_M$) was routinely measured from slug individuals using PAM-2000 fluorometer (Heinz Walz GmbH) after minimum of 20 min darkness. The measurements were carried out by placing a dark acclimated slug on to the side of an empty Petri dish and then pipetting all ASW media out, leaving the slug relatively immobile for the time required for the measurement. The light guide of PAM-2000 was hand-held at a ~ 45° angle respective to the slug, using the side and bottom of the Petri dish as support, and a saturating pulse was fired. PAM-

2000 settings used for $F_V/F_M$ measurements from the slugs were as follows: MB intensity 10 (maximum), MB frequency 0.6 kHz, high MB frequency 20 kHz (automatically on during actinic light illumination), MT pulse intensity 10 (maximum, PPFD > 10,000 µmol m$^{-2}$s$^{-1}$), MT pulse duration 0.8 s.

Measuring rapid light curves (RLCs) requires total immobilization of the slugs, a topic that has been thoroughly discussed by *Cruz et al., 2012*. Instead of using the anaesthetic immobilization technique described by *Cruz et al., 2012*, we tested yet another immobilization method to broaden the toolkit available for studying photosynthesis in Sacoglossan sea slugs. Alginate is a porous, biologically inert and transparent polymer that is widely used for fixing unicellular algae and cyanobacteria to create uniform and easy to handle biofilms or beads in for example biofuel research (*Kosourov and Seibert, 2009*; *Antal et al., 2014*). For the fixation of the slugs, an individual slug (representing one biological replicate) was placed on a Petri dish and a small drop of 1% alginate (m/v in H$_2$O) was pipetted on top of the slug, covering the slug entirely. Next, roughly the same volume of 0.5 mM CaCl$_2$ was distributed evenly to the alginate drop to allow the Ca$^{2+}$ ions to rapidly polymerize the alginate. The polymerization was allowed to continue for 10–30 s until the alginate had visibly solidified. All leftover CaCl$_2$ was removed with a tissue, and the slug fixed inside the alginate drop was placed under the fixed light guide of PAM-2000, in direct contact and in a 90° angle, for the measurement. After the measurement was over, the alginate drop was covered with abundant 1M Na-EDTA to rapidly chelate the Ca$^{2+}$ ions and depolymerize the alginate. Once the slug was visibly free of alginate, it was immediately transferred to fresh ASW for rinsing with a Pasteur pipette. The slugs usually recovered full movement, defined as climbing the walls of the container, in 10–20 min. The slugs were placed into a new tank for breeding purposes once motility had been restored. We also tested the effect of alginate fixation on $F_V/F_M$ during a 10-min time period, a typical length for RLC measurements, and no effect was noticeable (*Figure 5—figure supplement 1*). All RLCs from algae were measured from five to ten cells/measurement (representing one biological replicate), using otherwise the same setup as with the slugs, except that alginate fixation was not applied. The basic settings for RLC measurements were the same as with $F_V/F_M$ measurements, except for the MB intensity, which was adjusted to setting five with the algae to avoid oversaturation of the signal. Each light step lasted 90 s and the PPFDs that were used are shown in the figures. rETR was calculated as 0.42*Y(II)*PPFD, where 0.42 represents the fraction of incident photons absorbed by PSII, based on higher plant leaf assumptions, and Y(II) represents effective quantum yield of PSII photochemistry under illumination. NPQ was calculated as $F_M/F_{M'}-1$, where $F_{M'}$ represent maximum chlorophyll fluorescence of illuminated samples. See *Kalaji et al., 2014* for detailed descriptions of rETR and NPQ.

## Feeding experiments

In order to ensure that the slugs incorporate only specifically acclimated chloroplasts inside their own cells, the first feeding experiment was done with slug individuals that had been kept away from their food for 4 weeks in 5 l tanks filled with fresh ASW medium in their normal culture conditions. The coloration of the slugs was pale after the starvation period, indicating a decrease in the chloroplast content within the slugs. Altogether 107 starved slug individuals were selected for the feeding experiments and divided to two tanks filled with abundant *Acetabularia* in f/2 culture medium, one tank containing high-CO$_2$ acclimated algae (54 slugs) and the other one algae grown in ambient air (53 slugs). The tanks with the slugs and algae in them were put to their respective growth conditions for 4 days to allow the slugs to incorporate new chloroplasts inside their cells. The tanks were not aerated during the feeding, but the tank lids were unlocked for both feeding groups. The elevated CO$_2$ level in the closed culture cabinet posed a problem, as it noticeably affected the slug behavior by making them sessile in comparison to normal growth conditions, probably due to increased replacement of O$_2$ by CO$_2$. Because of this, the slugs selected for feeding on the high-CO$_2$ acclimated algae were fed in cycles where the tanks were in the closed CO$_2$ cabinet for most of the time during the daylight hours, but taken out every few hours and mixed with ambient air by stirring and kept in the ambient-air conditions for 1–2 hr before taking the tanks back to the high-CO$_2$ cabinet. The tanks were always left inside the closed cabinets for the nights in order to inflict minimal changes to the acclimation state of the algae. After 4 days of feeding, 50 slug individuals of similar size and coloration were selected from both feeding groups and distributed into 2 new 5 l tanks/feeding group, filled with approximately 2 l of 3.7% ASW medium and containing no algae. All slugs (25+25 ambient-air slugs, 25+25 high-CO$_2$ slugs) were moved to ambient-air growth conditions and

kept in the dark overnight in order to allow maximal incorporation of the chloroplasts before starting the starvation experiment in steady-light conditions.

For the second feeding experiment, the 4-week pre-starvation period of the slugs was discarded to see whether the differences in photosynthetic parameters between the two feeding groups could be inflicted just by allowing the slugs to replace their old kleptoplasts with the specific chloroplasts fed to them. We selected 100 slugs from normal growth conditions and divided them once again into two tanks containing f/2 culture medium and *Acetabularia* that had been acclimated to ambient air (50 slugs) or high $CO_2$ (50 slugs). The slugs were allowed to feed for 6 days, but otherwise the feeding protocol was identical to the one used in the first feeding experiment. After the sixth day of feeding, 45 slug individuals of similar size and coloration were selected from both feeding groups and divided to two new 5 l tanks (20+25 ambient-air slugs, 20+25 1% $CO_2$ slugs) filled with approximately 2 l of ASW and kept overnight in the dark before starting the starvation experiment in fluctuating light.

A third feeding experiment was conducted in order to create the red morphotype of *E. timida* (*González-Wangüemert et al., 2006*; *Costa et al., 2012*). Slug individuals from normal growth conditions were selected and placed in Petri dishes filled with f/2 culture medium and abundant red *Acetabularia*. The slugs were allowed to eat the algae for 2 days in the normal culture conditions of the slugs prior to the measurements. Three Petri dishes were filled with just the red form *Acetabularia* in f/2 culture medium in exactly the same conditions, and these algae were used for measurements regarding the red morphotypes of *Acetabularia* and *E. timida*.

## Starvation experiments

Two different starvation experiments were carried out with ambient-air slugs and high-$CO_2$ slugs. In the first one the slugs from the first feeding experiment were starved in steady-light conditions, where the only changes in the incident light were due to the day/night light cycle (12/12 hr). Here, all four tanks (25+25 ambient-air slugs, 25+25 high-$CO_2$ slugs) were placed under white LED lights (Växer PAR30 E27, 10 W; Ikea, Delft, The Netherlands; see *Figure 7—figure supplement 1B* for the spectrum) adjusted to PPFD 40 μmol $m^{-2}s^{-1}$. Temperature was maintained at 23°C and the tanks were not aerated during the starvation experiment apart from the passive gas flux that was facilitated by the unlocked lids of the tanks. Fresh ASW medium (approximately 500 ml) was added to the tanks every second day and the slugs were placed into new tanks with fresh ASW 1–2 times a week throughout the entire starvation period of 46 days. The day following the overnight dark period that the slugs were subjected to after the feeding experiment was noted as day 0 in the starvation experiments. $F_V/F_M$ during starvation was measured from individual slugs (representing one biological replicate) as indicated in the main text figures. Sampling caused losses to the slug populations on days 0, 5, and 15, when 10 slug individuals/group were selected for P700 oxidation kinetics measurements. Unfortunately, the P700$^+$ signals after 15 days in starvation were too weak and noisy for any meaningful interpretations. Before day 25, starvation induced mortality of both groups was 0. After that the ambient-air slug population suffered losses on days 27 (1 slug), 35 (2 slugs), 45 (5 slugs), 46 (1 slug) altogether nine slugs. For the high-$CO_2$ slug population the losses were as follows: day 27 (1 slug), 29 (1 slug), 31 (1 slug), 45 (8 slugs) and 46 (1 slug), 12 slugs in total. Lengths of the slugs were estimated from images taken at set intervals essentially as described by *Christa et al., 2018*. Images were taken with a cropped sensor DSLR camera equipped with a macro lens (Canon EOS 7D MKII + Canon EF-S 60 mm f/2.8 Macro lens; Canon Inc, Tokyo, Japan) and the body length of each slug individual was estimated using the image analysis software Fiji (*Schindelin et al., 2012*). The slugs from both feeding groups were pooled into one tank/group after day 25 in starvation.

The second starvation experiment was carried out using ambient-air slugs and high-$CO_2$ slugs from the second feeding experiment. The four tanks from both feeding groups (20+25 ambient-air slugs, 20+25 high-$CO_2$ slugs) were placed under a fluctuating light regime. The day/night cycle was maintained at 12/12 hr, but during the daylight hours the background illumination (PPFD 40 μmol $m^{-2}s^{-1}$) was supplemented with a 10 s pulse of high light (PPFD 1500 μmol $m^{-2}s^{-1}$) every 10 min. Both the background illumination and the high-light pulses originated from a programmable Heliospectra L4A greenhouse lamp (model 001.010; Heliospectra, Göteborg, Sweden; see *Figure 7—figure supplement 1B* for the irradiance spectra). All other conditions and procedures were identical to the ones used in the first starvation experiment. $F_V/F_M$ during starvation was measured as indicated in the main text figures. Sampling caused losses to the slug populations on days 0 and 6,

when 10 slugs/group were selected for P700 redox kinetics measurements, and on day 10, when five slugs/group were selected for RLC measurements. No images were taken during the starvation experiment in fluctuating light. The slugs were pooled into one tank/feeding group after 25 days in starvation. Chlorophyll was extracted from the slugs with N,N-dimethylformamide and chlorophyll *a/b* was estimated spectrophotometrically according to *Porra et al., 1989*.

## Acknowledgements

This study was funded by Academy of Finland (grants 307335 and 333421). VH was supported by Finnish Cultural Foundation, Väisälä Fund and University of Turku Graduate School (UTUGS). Sven. B Gould and his research group are acknowledged for the invaluable help in establishing laboratory cultures of both *E. timida* and *Acetabularia*, and Taras Antal for help with fluorescence induction measurements. We are grateful to associate professor Yagut Allahverdiyeva-Rinne for the FLVB antibody, and Martina Jokel-Toivanen is thanked for advice concerning its use.

## Additional information

### Funding

| Funder | Grant reference number | Author |
| --- | --- | --- |
| Academy of Finland | 307335 | Esa Tyystjärvi |
| Suomen Kulttuurirahasto | Graduate student grant | Vesa Havurinne |
| Suomalainen Tiedeakatemia | Graduate student grant | Vesa Havurinne |
| University of Turku graduate school, DPMLS | Graduate student grant | Vesa Havurinne |
| Academy of Finland | 333421 | Esa Tyystjärvi |

The funders had no role in study design, data collection and interpretation, or the decision to submit the work for publication.

### Author contributions

Vesa Havurinne, Conceptualization, Resources, Formal analysis, Funding acquisition, Validation, Investigation, Visualization, Methodology, Writing - original draft, Project administration; Esa Tyystjärvi, Conceptualization, Resources, Supervision, Funding acquisition, Validation, Project administration, Writing - review and editing

### Author ORCIDs

Vesa Havurinne http://orcid.org/0000-0001-5213-0905
Esa Tyystjärvi https://orcid.org/0000-0001-6808-7470

### Ethics

Animal experimentation: This study was performed in accordance with EU legislation and directives concerning scientific research on non-cephalopod invertebrates.

### Decision letter and Author response

Decision letter https://doi.org/10.7554/eLife.57389.sa1
Author response https://doi.org/10.7554/eLife.57389.sa2

## Additional files

### Supplementary files

- Transparent reporting form

## Data availability

All data generated or analysed during this study are included in the manuscript and supporting files. Source data files have been provided for Figures 2, 3, 4, Figure 4—figure supplement 1 and Figures 5, 6, 7B-E and 8.

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
