## [Decision Letter]

**Acceptance summary:**

The editors applaud your far-reaching investigation of chloroplast physiology in *Acetabularia* algae and upon their integration into sea slug cells. The study raises new hypotheses and provides strong evidence for putative mechanisms that explain the long-term maintenance of functional kleptoplasts inside animal cells, shedding light on a fascinating biological phenomenon.

**Decision letter after peer review:**

Thank you for submitting your article "Photosynthetic sea slugs inflict protective changes to the light reactions of the chloroplasts they steal from algae" for consideration by *eLife*. Your article has been reviewed by three peer reviewers, and the evaluation has been overseen by Christian Hardtke as the Senior and Reviewing Editor.

The reviewers have discussed the reviews with one another and the Reviewing Editor has drafted this decision to help you prepare a revised submission.

The reviewers all appreciate your manuscript, which aims to explore the mechanistic basis of a unique biological phenomenon. However, they are also of the opinion that your paper is too speculative at this point and requires further experimental evidence to substantiate your claims.

In particular, it is essential that you provide definite proof for the (elevated) presence of flavodiiron (FLV) in the plastids/slug tissue. The FLV hypothesis is crucial to your paper because it might explain protective P700 re-oxidation under fluctuating light. What is clearly needed then is firm evidence for the existence of an FLV in either the algae or the slug. For example, to start with, does the *Acetabularia*/slug genome encode FLVs? Or ideally, could an FLV be detected in the slug body or the algae?

Please note that data supporting the existence of FLVs in your study system are required for further consideration of your manuscript. In a revised version, please also address the individual review comments, which you can find pasted below.

Reviewer #1:

Sacoglossan sea slugs (*Elysia timida* in this study) are probably the only known animals capable of photosynthesis. The slugs eat certain algae (*Acetabularia acetabulum* in this study), extract (or rather "steal) their chloroplasts and incorporate them in their own cells. This phenomenon known as kleptoplasty allows Elysia to survive for longer under starvation conditions. However, inside the slug the chloroplasts are cut off from their normal host organism *Acetabularia*. As such they must rely on what they had when they were stolen. Therefore, the question is how the chloroplasts are able to survive and remain functional for a month or so inside the slug cells.

This study suggests that the chloroplasts inside the slug undergo a number of adaptations that may be life prolonging outside the host organism:

– A more oxidized plastoquinone pool, that would relieve oxidative stress on Photosystem II

– Rapid acidification of the thylakoid lumen and enhanced non-photochemical quenching (NPQ), a mechanism that can protect photosynthetic organisms from the effects of excess light.

– A high P700 oxidation capacity that may increase the longevity of chloroplasts in the slugs under fluctuating light. The high P700 oxidation capacity was attributed to the hypothetical presence of flavodiiron proteins (FLVs) as electron acceptors at the level of Photosystem I. The FLVs would have to originate from the *Acetabularia* algae.

This is a fascinating study that should be suitable for the journal and its broad audience. The manuscript is expertly written and the figures are of excellent quality.

It is believable that the above-mentioned photosynthetic adaptations may contribute to the longevity of the chloroplasts inside the slug cells. Overall, the data appear solid but they also raise some questions.

It is imaginable that the above-mentioned photosynthetic adaptations may contribute to the longevity of the chloroplasts inside the slug cells. Overall, the data are intriguing but also raise a number of questions, mostly with regard to the comparison of the algae and slug systems:

1) The flavodiiron (FLV) hypothesis is crucial to this paper and might explain protective P700 re-oxidation under fluctuating light. What is clearly needed here (and should be added) is evidence for the existence of a FLV in either the algae or the slug. Does the *Acetabularia* genome (or that of related species) encode FLVs? Or could an FLV be detected somehow (Western blot, mass spec) in the slug body or the algae?

2) With regard to the measurements: how does the animal body and physiology influence the photosynthetic parameters? Could altered CO2 concentrations or the addition of DCMU alter slug physiology or metabolism in a way that would be reflected at the level of slug photosynthetic parameters? Are the chlorophyll concentrations and the extinction properties of the algae and slug samples comparable so that equal light intensities would result in a similar level of chlorophyll excitation and therefore induce similar NPQ?

3)The electrochromic shift in Figure 4A was measured at 520 and 550 nm: is there any possible contribution here by electrochromism occurring in the animal body and outside the kleptoplast? It may be interesting to test this in kleptoplast-free Elysia slugs.

Reviewer #2:

This study provides a far-reaching investigation of chloroplasts physiology in *Acetabularia* algae and upon integration into sea slug cells. It raises new hypothesis and show strong evidences, for mechanisms explaining long-term maintenance of functional kleptoplasts inside the animal cells.

The Introduction is clear and well written, providing all necessary background to contextualize the study. The level of detail in the Materials and methods is impressive and incredibly useful. The results are explained in detail.

Overall, as a reader, I was left with a feeling of honest and clear explanations throughout the manuscript, as all scientific manuscript should be. Criticism to other studies is also made in a constructive manner.

I do not hesitate recommending this work for publication. It will be of interest for both specific communities investigating kleptoplasty (in Sacoglossa and others) but also for researchers investigating plant physiology and endosymbiosis theory.

Reviewer #3:

This manuscript describes spectroscopic investigation of kleptoplastids in the sea slug *Elysia timida*. The authors appear to have achieved a technical advance in establishing an experimental system for culturing populations of Elysia (and *Acetabulariaacetabulum*, a green alga that it feeds on), and this enables them to make extensive measurements of chlorophyll fluorescence and absorbance changes to investigate the photosynthetic electron transport chain in kleptoplastids in situ. The results suggest some changes in electron transport, oxidation of the PQ pool, and possible engagement of flavodiiron (FLV) proteins in Elysia vs. *Acetabularia*, but overall the data are neither conclusive nor convincing. How Elysia is able to maintain functional photosynthetic electron transfer in kleptoplasts is an interesting and longstanding question, but unfortunately this manuscript does not provide a clear answer.

1) The authors assert that they "optimized a completely new set of biophysical methods", but these all appear to be well established methods that are just now being used with the sea slugs.

2) Observations of a wave-like chlorophyll fluorescence decay (Figure 2D) are discussed relative to previous results in *Chlamydomonas* in which a role for a type II NDH (NDA2) was invoked. However, beyond a superficial similarity in phenotype, no further experimental evidence (e.g. inhibition with a type II NDH inhibitor) is presented to support this speculation.

3) Some relatively small differences in P700 redox kinetics between Elysia and *Acetabularia* grown in ambient vs. high CO_2_ (Figures 4 and 5) are interpreted as differences in FLVs as electron sinks, but without any direct measurement of FLV protein levels by immunoblotting or mass spec proteomics. This analysis is not at all convincing, and it appears to be based on the previous observation that "high CO_2_ induces downregulation of certain FLVs in cyanobacteria”. The situation could be completely different in *Acetabularia*, which is a green alga (not a cyanobacterium), and even the authors admit that the differences could be due to "other changes caused by the high-CO_2_ acclimation". The authors have not ruled out other possible interpretations, such as differences in the Mehler reaction or even PTOX-catalyzed alternative electron transport, either of which could result in the observed differences in P700 redox kinetics.

4) The data in Figure 7A show a slight difference in loss of PSII activity in fluctuating light between Elysia with ambient air vs. high CO_2_ kleptoplasts. This difference is largely attributable to differences in F_M_ during the period of 10-25 days after starvation – this seems inconsistent with the proposed effects on PSII repair (subsection “High P700 oxidation capacity improves kleptoplast longevity under fluctuating light”), which would be expected to cause an increase in F_0_ instead. The caveat mentioned in the second paragraph of the subsection “P700 redox kinetics in *E. timida* are affected by the acclimation status of its prey”, also applies to this experiment.

5) Overall, the conclusions of most of the presented experiments are tentative, and this is reflected in the language used by the authors throughout the manuscript (e.g. "could" and "probably"). The more definitive statement at the end of the Introduction ("It is also clear that PSI utilizes oxygen sensitive flavodiiron proteins (FLVs) as alternative electron sinks in both the slugs and the algae, and this sink protects the photosynthetic apparatus from light-induced damage in *E. timida*."), which summarizes the authors' main conclusion, is not justified by the data in this manuscript.

---

## [Author Response]

The reviewers all appreciate your manuscript, which aims to explore the mechanistic basis of a unique biological phenomenon. However, they are also of the opinion that your paper is too speculative at this point and requires further experimental evidence to substantiate your claims.In particular, it is essential that you provide definite proof for the (elevated) presence of flavodiiron (FLV) in the plastids/slug tissue. The FLV hypothesis is crucial to your paper because it might explain protective P700 re-oxidation under fluctuating light. What is clearly needed then is firm evidence for the existence of an FLV in either the algae or the slug. For example, to start with, does the Acetabularia/slug genome encode FLVs? Or ideally, could an FLV be detected in the slug body or the algae?

We now provide a Western blot showing that in *Acetabularia acetabulum* a protein band of appropriate size reacts with an antibody that recognizes FLVA and FLVB proteins of *Chlamydomonas reinhardtii* (Figure 6 and associated text, Results subsection “P700 redox kinetics in *E. timida* are affected by the acclimation status of its prey”, last paragraph, Discussion subsections “Flavodiiron proteins function as alternative electron sinks in *E. timida* and *Acetabularia*” and “High P700 oxidation capacity improves kleptoplast longevity under fluctuating light, Materials and methods subsection “Protein analysis”).

Please note that data supporting the existence of FLVs in your study system are required for further consideration of your manuscript. In a revised version, please also address the individual review comments, which you can find pasted below.Reviewer #1:Sacoglossan sea slugs (Elysia timida in this study) are probably the only known animals capable of photosynthesis. The slugs eat certain algae (Acetabularia acetabulum in this study), extract (or rather "steal) their chloroplasts and incorporate them in their own cells. This phenomenon known as kleptoplasty allows Elysia to survive for longer under starvation conditions. However, inside the slug the chloroplasts are cut off from their normal host organism Acetabularia. As such they must rely on what they had when they were stolen. Therefore, the question is how the chloroplasts are able to survive and remain functional for a month or so inside the slug cells.This study suggests that the chloroplasts inside the slug undergo a number of adaptations that may be life prolonging outside the host organism:– A more oxidized plastoquinone pool, that would relieve oxidative stress on Photosystem II.– Rapid acidification of the thylakoid lumen and enhanced non-photochemical quenching (NPQ), a mechanism that can protect photosynthetic organisms from the effects of excess light.– A high P700 oxidation capacity that may increase the longevity of chloroplasts in the slugs under fluctuating light. The high P700 oxidation capacity was attributed to the hypothetical presence of flavodiiron proteins (FLVs) as electron acceptors at the level of Photosystem I. The FLVs would have to originate from the Acetabularia algae.This is a fascinating study that should be suitable for the journal and its broad audience. The manuscript is expertly written and the figures are of excellent quality.It is believable that the above-mentioned photosynthetic adaptations may contribute to the longevity of the chloroplasts inside the slug cells. Overall, the data appear solid but they also raise some questions.It is imaginable that the above-mentioned photosynthetic adaptations may contribute to the longevity of the chloroplasts inside the slug cells. Overall, the data are intriguing but also raise a number of questions, mostly with regard to the comparison of the algae and slug systems:1) The flavodiiron (FLV) hypothesis is crucial to this paper and might explain protective P700 re-oxidation under fluctuating light. What is clearly needed here (and should be added) is evidence for the existence of a FLV in either the algae or the slug. Does the Acetabularia genome (or that of related species) encode FLVs? Or could an FLV be detected somehow (Western blot, mass spec) in the slug body or the algae?

The existence of FLVs in *Acetabularia* was confirmed by Western blot analysis using an antibody raised against *Chlamydomonas reinhardtii* FLVB protein; the antibody reacts with FLVA and FLVB of *C. reinhardtii* (Figure 6 and associated text, Results subsection “P700 redox kinetics in *E. timida* are affected by the acclimation status of its prey”, last paragraph, Discussion subsections “Flavodiiron proteins function as alternative electron sinks in *E. timida* and *Acetabularia*” and “High P700 oxidation capacity improves kleptoplast longevity under fluctuating light”, Materials and methods subsection “Protein analysis”).

2) With regard to the measurements: how does the animal body and physiology influence the photosynthetic parameters? Could altered CO_2_ concentrations or the addition of DCMU alter slug physiology or metabolism in a way that would be reflected at the level of slug photosynthetic parameters?

This in an important point, and unfortunately we do not have a definitive answer for the effects of DCMU and elevated CO_2_ levels on slug metabolism. Based on our observations about the viability of the slugs, however, they do not seem to be severely affected by DCMU when used in the 10 µM concentration utilized in this study. Also a previous study has shown that another PSII blocking diuron, monolinuron, does not dramatically affect viability of another slug species, *Plakobranchus ocellatus* (Christa et al. 2013 Plastid-bearing sea slugs fix CO_2_ in the light but do not require photosynthesis to survive. Proc. Biol. Sci. 281:20132493. DOI: 10.1098/rspb.2013.2493).

Based on our own observations, *E. timida* slugs do bleach fast when starved in the light in the presence of DCMU, but in our study only freshly fed slugs were utilized in DCMU experiments. Since the slugs do not e.g. curl up when exposed to DCMU, we believe that the optical properties of the slugs are not severely affected by DCMU treatment, when compared to non-treated slug individuals.

As for elevated CO_2_ levels, they do seem to affect the motility of the slugs, as they become sessile, as described in the Materials and methods of the text. However, all slug measurements from high CO_2_ slugs were performed after at least an overnight settling period in ambient air, after which the slugs had recovered full motility. The possibility of DCMU not blocking all PSII units in the slugs is now discussed in the third paragraph of the subsection “Full photochemical reduction of the electron transfer chain during a dark-to-light transition is delayed in *E. timida*”.

Are the chlorophyll concentrations and the extinction properties of the algae and slug samples comparable so that equal light intensities would result in a similar level of chlorophyll excitation and therefore induce similar NPQ?

The optical properties between the slugs and the algae do differ. However, our unpublished data related to another study show that the slugs in fact absorb less light on a Chl basis than the algae. This affects light utilization by photosynthesis, i.e. less photons are utilized by the slugs than the algae. We have tried to downplay this problem by mainly analyzing data obtained by truly saturating pulses (PPFD 10 000 µmol m^-2^s^-1^ or more). When this was not possible, as was the case with Aquapen used for the OJIP measurements, we felt it necessary to include data obtained with a set of high light pulses of differing intensities that showcase the saturating effect of the light pulse that was referred to in the main text.

It is certainly true that since the light absorption properties of the slugs are different from the algae, the light intensities presented in the X-axes of the RLC figures do not tell the whole truth concerning light utilization by the slugs vs. the algae. However, since the slugs absorb less light per Chl, the stronger NPQ in the slugs is even more remarkable, further emphasizing our point, that the slugs induce physiological changes in the chloroplasts that help the chloroplasts remain functional inside the slug tissue. As the data showing the absorption on a Chl basis has not been published yet, we cannot discuss this point in this manuscript in full detail, but changes have been made to the text that emphasize the optical differences between the slugs and the algae (subsection “Full photochemical reduction of the electron transfer chain during a dark-to-light transition is delayed in *E. timida*”, third paragraph).

3)The electrochromic shift in Figure 4A was measured at 520 and 550 nm: is there any possible contribution here by electrochromism occurring in the animal body and outside the kleptoplast? It may be interesting to test this in kleptoplast-free Elysia slugs.

ECS and also P700^+^ measurements from kleptoplast-free slugs have been added (Figure 4—figure supplement 1 and associated text, subsections “Chloroplasts in *E. timida* exhibit strong build-up of proton motive force in the light”and “*E. timida* and *Acetabularia* utilize oxygen dependent electron sinks from PSI”, Materials and methods subsection “Organisms and culture conditions”, last paragraph).

Reviewer #3:This manuscript describes spectroscopic investigation of kleptoplastids in the sea slug Elysia timida. The authors appear to have achieved a technical advance in establishing an experimental system for culturing populations of Elysia (and Acetabularia acetabulum, a green alga that it feeds on), and this enables them to make extensive measurements of chlorophyll fluorescence and absorbance changes to investigate the photosynthetic electron transport chain in kleptoplastids in situ. The results suggest some changes in electron transport, oxidation of the PQ pool, and possible engagement of flavodiiron (FLV) proteins in Elysia vs. Acetabularia, but overall the data are neither conclusive nor convincing. How Elysia is able to maintain functional photosynthetic electron transfer in kleptoplasts is an interesting and longstanding question, but unfortunately this manuscript does not provide a clear answer.1) The authors assert that they "optimized a completely new set of biophysical methods", but these all appear to be well established methods that are just now being used with the sea slugs.

The word “completely” has been taken out.

2) Observations of a wave-like chlorophyll fluorescence decay (Figure 2D) are discussed relative to previous results in Chlamydomonas in which a role for a type II NDH (NDA2) was invoked. However, beyond a superficial similarity in phenotype, no further experimental evidence (e.g. inhibition with a type II NDH inhibitor) is presented to support this speculation.

We find the reviewer’s criticism justified. Unfortunately, creating a new batch of red *Acetabularia* was not possible. We are therefore unable to provide further proof for the involvement of NDA2 in the wave-phenomenon in *Acetabularia* and now we emphasize the need for further proof (subsection “Weakened dark reduction of the PQ pool lowers electron pressure in *E. timida* kleptoplasts”, second paragraph).

3) Some relatively small differences in P700 redox kinetics between Elysia and Acetabularia grown in ambient vs. high CO_2_ (Figures 4 and 5) are interpreted as differences in FLVs as electron sinks, but without any direct measurement of FLV protein levels by immunoblotting or mass spec proteomics. This analysis is not at all convincing, and it appears to be based on the previous observation that "high CO_2_ induces downregulation of certain FLVs in cyanobacteria”. The situation could be completely different in Acetabularia, which is a green alga (not a cyanobacterium), and even the authors admit that the differences could be due to "other changes caused by the high-CO_2_ acclimation". The authors have not ruled out other possible interpretations, such as differences in the Mehler reaction or even PTOX-catalyzed alternative electron transport, either of which could result in the observed differences in P700 redox kinetics.

The existence of FLVs in *Acetabularia* has now been shown by Western blot analysis (Figure 6 and associated text, Results subsection “P700 redox kinetics in *E. timida* are affected by the acclimation status of its prey”, last paragraph, Discussion subsections “Flavodiiron proteins function as alternative electron sinks in *E. timida* and *Acetabularia*” and “High P700 oxidation capacity improves kleptoplast longevity under fluctuating light”, Materials and methods subsection “Protein analysis”). However, we did not notice signs of FLV downregulation in high CO_2_*Acetabularia*, even though this has in fact been noted in another green alga (Jokel et al., 2015; the reference has been added and discussed in the text). The entire manuscript has been altered to take into account the new results that do not show changes in FLV proteins when the algae are exposed to high CO_2_, with discussion covering the alternative explanations suggested by the reviewer (subsection “High P700 oxidation capacity improves kleptoplast longevity under fluctuating light”, first paragraph).

4) The data in Figure 7A show a slight difference in loss of PSII activity in fluctuating light between Elysia with ambient air vs. high CO_2_ kleptoplasts. This difference is largely attributable to differences in F_M_ during the period of 10-25 days after starvation – this seems inconsistent with the proposed effects on PSII repair (subsection “High P700 oxidation capacity improves kleptoplast longevity under fluctuating light”), which would be expected to cause an increase in F_0_ instead. The caveat mentioned in the second paragraph of the subsection “P700 redox kinetics in E. timida are affected by the acclimation status of its prey”, also applies to this experiment.

We agree that photodamage to PSII, if it occurred without side effects, would be expected to lead to an increase of F_0_ with no effect on F_M_. However, photodamage to PSII has been studied for decades and usually F_M_ decreases but F_0_ stays almost unchanged. Increase in F_0_ has been seen in a few cases, especially under anaerobic conditions (see the review by Tyystjärvi 2008, Coord Chem Rev 252: 361-376) or in very low light (see Tyystjärvi and Aro 1996, PNAS 93: 2213-2218). An obvious reason for an unchanged F_0_ is that the photoinhibited PSII reaction centers quench excitation energy. In higher plants, this quenching behavior has also been experimentally verified (Matsubara and Chow, 2004). The behavior of F_0_ during a photoinhibition experiment depends on the experimental details, as quenching by photoinhibited PSII lowers both F_M_ and F_0_ while photodamage would simultaneously cause an increase in F_0_. In the case of starving slugs, the overall decrease in the number of kleptoplasts adds a third factor to the equation. Due to these complications, the primary data about F_M_ and F_0_ remain descriptive and we base our conclusion on the faster decline of F_V_/F_M_ in high-CO_2_ slugs. F_V_/F_M_, according to the most basic interpretations of chlorophyll fluorescence data (stemming from the works of Stern and Volmer, and in photosynthesis of Warren Butler; for a review, see Porcar-Castell et al. 2014, J Exp Bot 65: 4065-4095) is insensitive to factors that similarly affect F_M_ and F_0_.

The above considerations are outside of the scope of the study, and therefore we only very shortly refer to the various factors that may have affected F_0_ and F_M_ during the experiment (subsection “High-CO_2_*E. timida* kleptoplasts are sensitive to fluctuating light”, second paragraph).

5) Overall, the conclusions of most of the presented experiments are tentative, and this is reflected in the language used by the authors throughout the manuscript (e.g. "could" and "probably"). The more definitive statement at the end of the Introduction ("It is also clear that PSI utilizes oxygen sensitive flavodiiron proteins (FLVs) as alternative electron sinks in both the slugs and the algae, and this sink protects the photosynthetic apparatus from light-induced damage in E. timida."), which summarizes the authors' main conclusion, is not justified by the data in this manuscript.

In the light of our new results, we have revised our manuscript, and less weight is put on FLVs as the only alternative electron sinks from PSI. However, we still believe that our data in conjunction with the current literature is enough to suggest that FLVs do function as electron sinks in *E. timida* and *Acetabularia.* We do not claim to have solved the mystery of chloroplast longevity in photosynthetic sea slugs, but our findings, supported by novel and strong data, indicate that the functionality of the chloroplasts is altered once inside the slugs and these changes can be photoprotective.